# Analysis of transcriptional changes in the immune system associated with pubertal development in a longitudinal cohort of children with asthma

Justyna A. Resztak [1,8], Jane Choe [1,8], Shreya Nirmalan [1], Julong Wei[1], Julian Bruinsma[2], Russell Houpt[2], Adnan Alazizi[1], Henriette E. Mair-Meijers[1], Xiaoquan Wen [3], Richard B. Slatcher[4], Samuele Zilioli[2,5], Roger Pique-Regi[1,6] ✉ & Francesca Luca [1,6,7] ✉

Puberty is an important developmental period marked by hormonal, metabolic and immune changes. Puberty also marks a shift in sex differences in susceptibility to asthma. Yet, little is known about the gene expression changes in immune cells that occur during pubertal development. Here we assess pubertal development and leukocyte gene expression in a longitudinal cohort of 251 children with asthma. We identify substantial gene expression changes associated with age and pubertal development. Gene expression changes between pre- and post-menarcheal females suggest a shift from predominantly innate to adaptive immunity. We show that genetic effects on gene expression change dynamically during pubertal development. Gene expression changes during puberty are correlated with gene expression changes associated with asthma and may explain sex differences in prevalence. Our results show that molecular data used to study the genetics of early onset diseases should consider pubertal development as an important factor that modifies the transcriptome.

Puberty is an important developmental period marked by hormonal, metabolic and immune changes, which have been implicated in disease predisposition later in life. Starting at puberty the physiology of men and women is influenced by different hormones - primarily androgens such as testosterone in males and estrogen and progesterone in females - with broad effects on all body systems[1]. The typical age for puberty onset is approximately 9–13.5 years in males and 8.5–13 years in females, yet there are fundamental differences between males and females in the dynamics of the reactivation of the gonadotropic axis at puberty onset[2]. Physiological changes during this period that are common to both sexes include growth spurts, increase in BMI[3], growth of body hair, skin changes (increased oil production and acne), and the maturation of gonads[4]. Females experience breast growth and menarche, while males undergo deepening of the voice and develop facial hair[4].

In addition to these physiological changes, certain pathological conditions also manifest during puberty or develop sex-specific symptom profiles[5–8]. In particular, incidence of many autoimmune

[1]Center for Molecular Medicine and Genetics, Wayne State University, Detroit, MI, USA. [2]Department of Psychology, Wayne State University, Detroit, MI, USA. [3]Department of Biostatistics, University of Michigan, Ann Arbor, MI, USA. [4]Department of Psychology, University of Georgia, Athens, GA, USA. [5]Department of Family Medicine and Public Health Sciences, Wayne State University, Detroit, MI, USA. [6]Department of Obstetrics and Gynecology, Wayne State University, Detroit, MI, USA. [7]Department of Biology, University of Rome Tor Vergata, Rome, Italy. [8]These authors contributed equally: Justyna Resztak, Jane Choe. ✉e-mail: rpique@wayne.edu; fluca@wayne.edu

diseases increases in the peri-pubertal period, with females bearing a higher disease risk than males[9–12]. Puberty also marks a shift in sex differences in susceptibility to asthma - males have higher asthma prevalence in childhood, but starting from young adulthood, females are more prone to asthma[13,14] Sex hormones are hypothesized to play a role in the switch of sex skew, as asthma symptoms are also known to worsen around menstruation[15,16]. Recently, a Mendelian randomization study showed a protective effect of increased sex hormone-binding globulin (SHBG) levels on asthma onset, with a larger effect in females than males[17]. Furthermore, in a cross-sectional study of a large cohort of children with severe asthma, circulating levels of androgens were found to positively associate with improved lung function and symptom control in pubescent males, while late puberty in females was associated with increased estradiol levels and decreased lung function[18] Despite sex differences in susceptibility to asthma before and after puberty onset[13], earlier onset of puberty has been found to increase the risk of asthma in both sexes[19]. Age at menarche is also known to have substantial effects on growth and disease[20]. Earlier menarche has been associated with increased risk of asthma[19], cardiovascular disease[21], type 2 diabetes[22] and higher BMI[20]. A thorough understanding of the longitudinal changes in the immune system that accompany puberty is needed to understand their impacts on asthma and other disease susceptibility.

The molecular mechanisms underlying the physiological and pathological changes in the immune system during puberty remain understudied and largely uncharacterized. Previous studies have explored the effects of puberty on DNA methylation[23–28] in peripheral blood immune cells. DNA methylation is an important epigenetic mark regulating gene expression during development and aging[29]. Hormonal changes in puberty have been associated with changes in DNA methylation[23]. For example, differentially methylated regions between pre- and post-pubertal females were enriched for estrogen response element, reproductive hormone signaling and immune and inflammatory responses, while differentially methylated regions in males were enriched for genes involved in adrenaline and noradrenaline biosynthesis[24]. Prior studies found a larger number of genomic regions changing methylation status during puberty in females than in males[24,30]. Furthermore, regions experiencing changes in methylation status in both sexes show changes with higher magnitude in females[26]. In contrast, other studies on pre- and post-pubertal subjects found limited differences in DNA methylation patterns between sexes[25,27]. These inconsistencies may be due to age differences between studies and could be further investigated by employing a study design that focuses on all pubertal developmental stages rather than comparing pre- and post-pubertal stages only.

Although DNA methylation findings support the hypothesis that blood gene expression during puberty undergoes reprogramming, little is known about the transcriptional changes accompanying human pubertal development, particularly from a genome-wide perspective. Most studies to date explored epigenetic differences between pre- and post-puberty, yet we don't have a complete picture of the gene expression changes during pubertal development in healthy or asthmatic children.

Here we analyze a longitudinal cohort of 103 females and 148 males aged 10-17 years old and diagnosed with childhood onset asthma. We investigate the associations between pubertal factors (i.e., age, pubertal stage, and menarche status in females), asthma symptoms, and genome-wide gene expression in blood. We identify gene expression changes in peri-pubertal males and females for thousands of genes over a short time period, as well as gene expression changes associated with pubertal development. Substantial transcriptional changes detected between pre- and post-menarcheal females suggest a shift from predominantly innate to adaptive immunity as females sexually matured. To assess the contribution of genetic factors, we explore the effects of genetic variation on gene expression changes

associated with puberty using an expression quantitative trait locus (eQTL) mapping approach, and show that genetic variation interacts with puberty to influence gene expression. Our work demonstrates that gene expression changes during puberty are correlated with gene expression changes associated with asthma and may explain differences in prevalence between males and females.

## Results
### Gene expression changes associated with age
Participants in this study were recruited as part of the longitudinal Asthma in the Lives of Families Today (ALOFT) study. The ALOFT study recruited males and females with childhood onset asthma and between 10 and 17 years old from Metropolitan Detroit. Children were followed annually for up to three years.

We collected gene expression, genotype, puberty and menarche data for a total of 251 participants. For 163 participants we collected data for two timepoints: baseline and follow-up (Fig. 1, Table S1). Average time to follow-up was 1.27 years (SD = 0.46). The entire range of puberty stages, measured using a scale equivalent to Tanner stages[31], were represented for both sexes.

We employed a longitudinal study design to investigate leukocyte gene expression changes in peri-pubertal males and females. We modeled gene expression with DESeq2[32] in each sex separately and identified 160 genes in females and 535 genes in males (Fig. 2a, Fig. S2, Supplementary Data 1, 2), whose expression changed between the two time points. Genes differentially expressed as females grew older were enriched among genes differentially expressed in males (OR = 9, $p$-value = $6.5 \times 10^{-16}$, Fig. 2b), suggesting a similar effect of age on gene expression in both sexes, despite the difference in the number of significant genes.

Gene expression changes associated with puberty are likely to be associated with age, but also with pubertal development and other developmental changes that happen during this period of life in children. To further validate these results, we considered all 251 individuals with at least one observation and used a cross-sectional study design to identify gene expression differences associated with the age of the donor. We observed a significant correlation between the gene expression changes measured longitudinally and cross-sectionally (Spearman rho = 0.43 in females, rho = 0.3 in males, $p$-value $< 2.2 \times 10^{-16}$), thus strengthening the evidence of gene expression changes associated with age in peri-pubertal children (Fig. 2c, d, Supplementary Data 3, 4).

To directly investigate whether the observed longitudinal changes in gene expression were similar in males and females, we used multivariate adaptive shrinkage (mash)[33]. This method uses correlations between effects measured in different conditions (e.g., males and females) to improve effect size estimates, increasing the power of discovery and enabling direct comparison between effects measured in different individuals. Applying this method to the results of the longitudinal analysis, we discovered over two thousand differentially expressed genes (Fig. 2e, Supplementary Data 5, 6). Specifically, we found 818 genes in males and 746 genes in females being upregulated when children grew older, and 1482 genes in males and 1439 genes in females being downregulated when children grew older. We found that these gene expression changes were largely shared between sexes, with 721 genes upregulated as children grew older and 1404 genes downregulated as children grew older in both sexes (Spearman rho = 0.95, $p$-value $< 2.2 \times 10^{-16}$, Fig. 2e, g), this could be because we have limited power to detect sex-specific effects of age and puberty with the current sample size. One of the genes whose expression increased dramatically as a function of age was *PLXND1* (Fig. 3a), which encodes a semaphorin receptor important for thymocyte development[34]. One of the genes most strongly downregulated as a function of age was *MACROD2* (Fig. 3b), which encodes a deacetylase involved in removing ADP-ribose from mono-ADP-ribosylated proteins. *MACROD2* deletions lead to

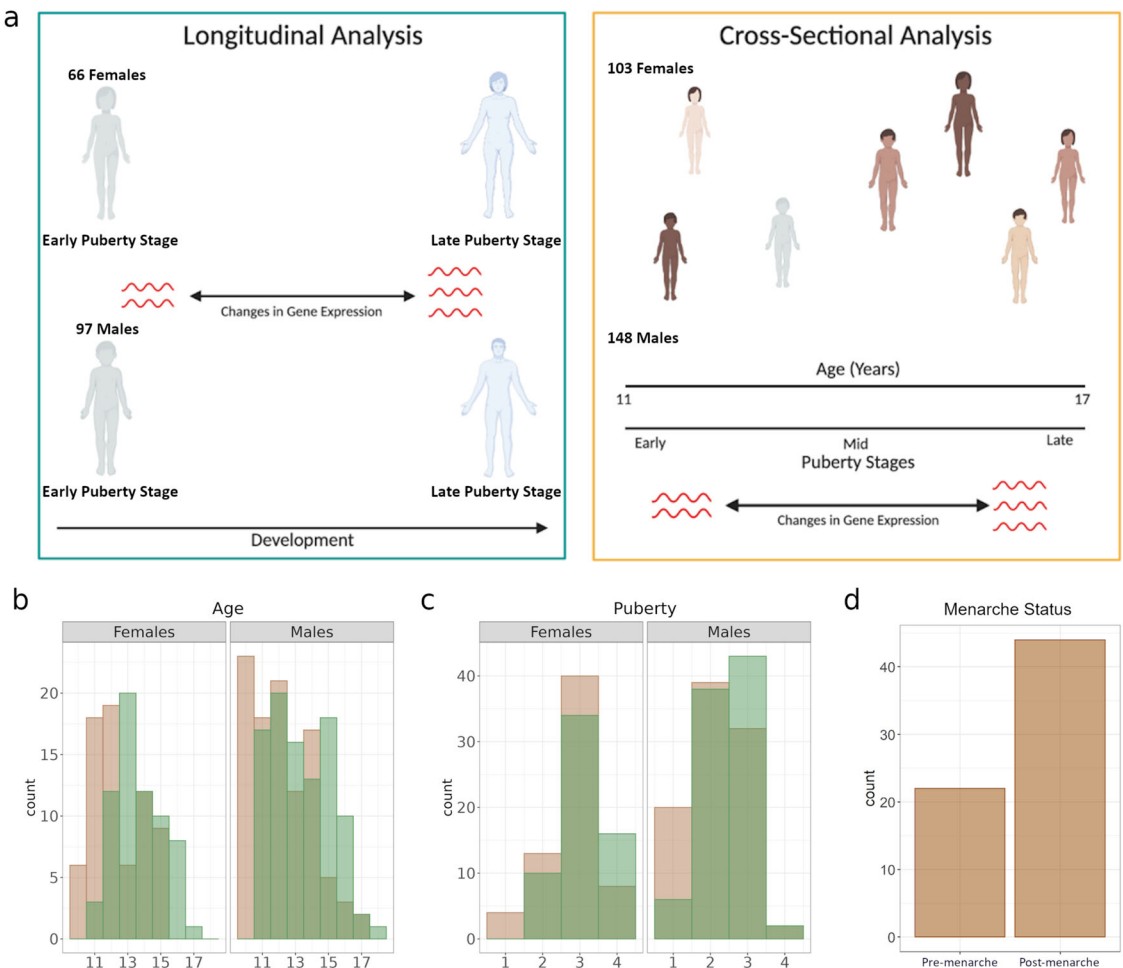

**Fig. 1 | Analysis of gene expression in peri-pubertal males and females. a** – Study design. We employed longitudinal and cross-sectional study designs to investigate the patterns of gene expression associated with pubertal development, **b** – Histograms of age distribution in females (left panel) and males (right panel) at baseline (tan) and at follow-up (green), **c** – Histograms of Pubertal Development Score[31] distribution in females (left panel) and males (right panel) at baseline (tan) and at follow-up (green), **d** - Bar graph of menarcheal status in the cross-sectional sample.

congenital anomalies of multiple organs[35]. *MACROD2* has been demonstrated to increase estrogen receptor coactivator binding to estrogen response elements[36], while its close paralog, *MACROD1*, is known to enhance transcriptional activation by androgen[37] and estrogen receptors[38].

There are limited data available on immune gene expression changes throughout the lifespan. A recent study investigated gene expression changes between the eighth and ninth decade of life[39], using a longitudinal study design. Here, we asked to what extent age effects on gene expression were similar at different life stages, i.e. during puberty and aging. Of the 1291 genes changing expression with age in the elderly, 188 were also differentially expressed in both sexes in our longitudinal peri-pubertal sample (OR = 1.2, *p*-value = 0.03). Among the genes differentially expressed in both youth and elderly, 44 genes increased expression and 76 decreased expression as a function of age in both groups. These genes may be unrelated to ageing and puberty, specifically; instead they may be associated with age-related biological processes shared across the lifespan. For 68 genes, age was associated with opposite effects on gene expression in peri-pubertal children and the elderly.

To understand what biological processes were represented by the observed gene expression changes in peri-pubertal children, we performed Gene Ontology and KEGG Pathways enrichment analyses on the longitudinal results. Genes downregulated with age in both sexes were enriched for genes in the Herpes simplex virus infection pathway

(*q*-value = 3.1 × 10⁻¹⁶). Genes upregulated when the children grew older were most strongly enriched for genes involved in biological processes related to carbohydrate metabolism (*q*-value = 1.7 × 10⁻⁴), neutrophil activation (*q*-value = 7.2 × 10⁻⁴) and vacuole organization (*q*-value = 4.5 × 10⁻⁴) (Fig. 3c)[40]. We also found significant enrichments for genes in the following biological processes and pathways: lysosome (*q*-value = 1.8 × 10⁻⁶), carbon metabolism (*q*-value = 0.01), and glycan degradation (*q*-value = 0.01) (Fig. 3d).

## Gene expression changes across pubertal stages

Despite the association between age and puberty, children of the same age might be at different stages of pubertal development[41]. To directly investigate transcriptional changes in immune cells during pubertal development, we used the same longitudinal study design described above and considered a quantitative pubertal development index analogous to Tanner stages[42,43] that ranged from 1 (no development) to 4 (completed development)[31] (Table S2). We found 108 genes differentially expressed with advancement through puberty in females and none in males (10% FDR) (Fig. 4a, Supplementary Data 7, 8). These results may reflect the different timing of puberty between females and males, with females in our study being more advanced in their pubertal development, compared to males. Among the 108 genes differentially expressed as females progressed through puberty, 72 of them were also differentially expressed as children became older: most genes (66) were downregulated, while only six genes were upregulated.

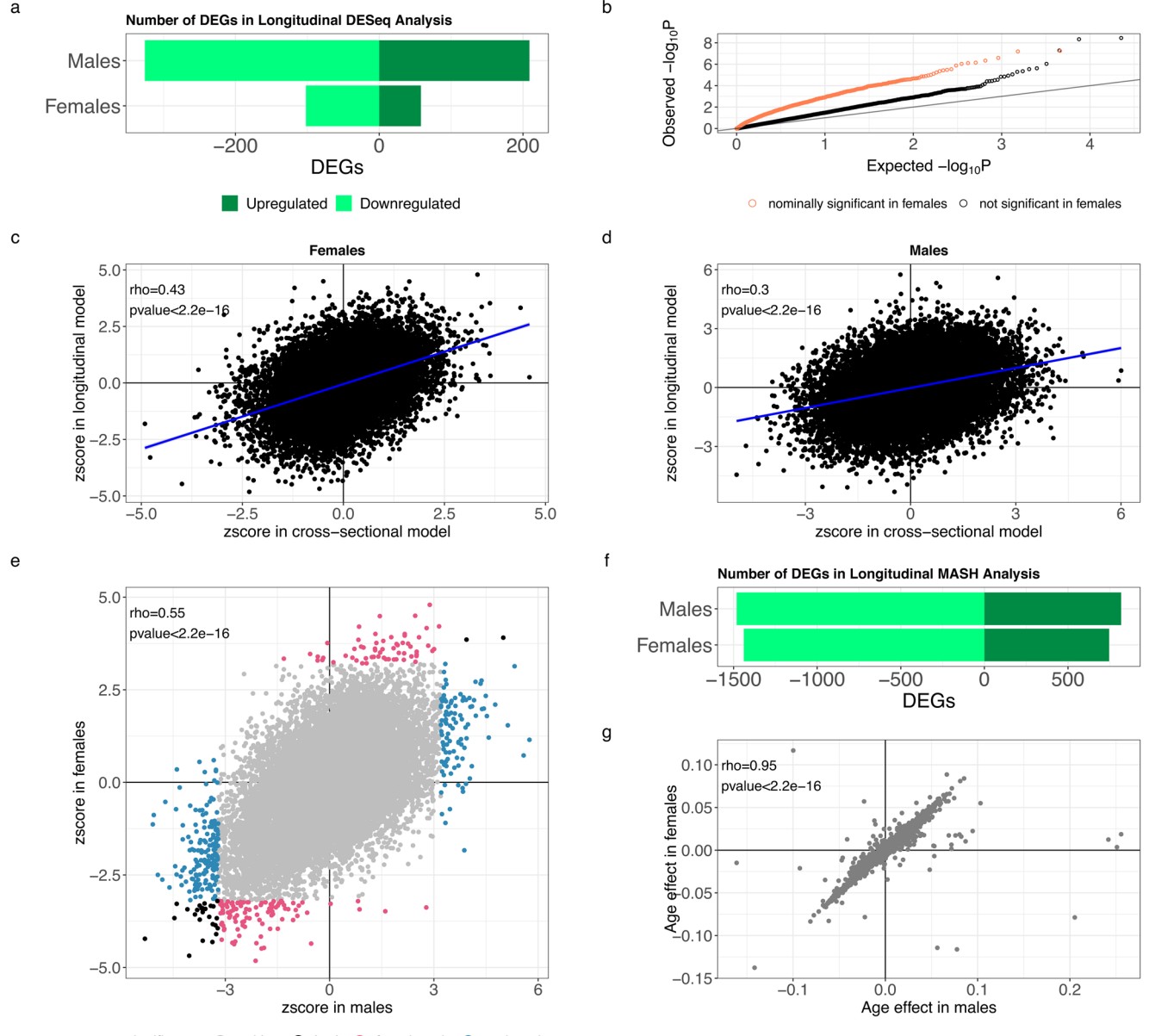

**Fig. 2 | Gene expression changes over time in peri-pubertal males and females.**
**a** Barplot with the number of genes differentially expressed (DEGs) with age in the two sexes (10% FDR), **b** QQplot of *p*-values of age effect on gene expression in males from DESeq2 two-sided Wald test, split by whether the gene was also nominally differentially expressed in females (*p*-value < 0.05, coral) or not significant (*p*-value > 0.05, black), **c**, **d** Scatterplot of z-scores of age effects on gene expression in females (**c**) and males (**d**) analyzed with the cross-sectional model (x axis) or longitudinal model (y axis). Correlation assessed with Spearman's Rho and its two-sided *p*-value. Blue line represents the linear regression trendline, **e** Scatterplot of normalized effect sizes (z-scores) from longitudinal differential gene expression analysis as males (x axis) and females (y axis) grew older. Color denotes significance in each analysis as indicated by the legend. Correlation assessed with Spearman's Rho and its two-sided *p*-value. **f** Barplot with the number of genes differentially expressed (DEGs) with age in the two sexes analyzed together with multivariate adaptive shrinkage (mash, 10% LFSR). Legend as in Fig. 2a, **g** Scatterplot of age effects on gene expression in males (x axis) and females (y axis) from mash analysis in both sexes jointly. Color denotes significance (10% LFSR) as indicated by the legend in Fig. 2e. Correlation assessed with Spearman's Rho and its two-sided *p*-value.

Our study design allowed us to ask whether similar gene expression changes were observed across individuals experiencing different stages of pubertal development at the time of sampling using the cross-sectional sample (Fig. S1). To this end, we considered all individuals with at least one observation in a cross-sectional study design in each sex. We observed significant correlations (Spearman rho = 0.13 in females and rho = 0.31 in males, *p*-value < 2.2 × 10$^{-16}$) between the gene expression log fold changes from the longitudinal and cross-sectional models (Supplementary Data 9, 10). Differences in the results between the two models could be due to additional sources of variation which

are usually present in cross-sectional datasets, including genetic and environmental effects that vary across individuals. Nevertheless, cross-sectional results support the findings of gene expression changes across pubertal stages from the longitudinal analysis.

To compare the transcriptional effects we discovered with previously-known epigenetic changes associated with pubertal development, we considered the overlap between genes differentially expressed as females advanced through puberty and 338 genes previously associated with regions differentially methylated in pre- and post-pubertal females[24]. Three of the 108 genes changing expression

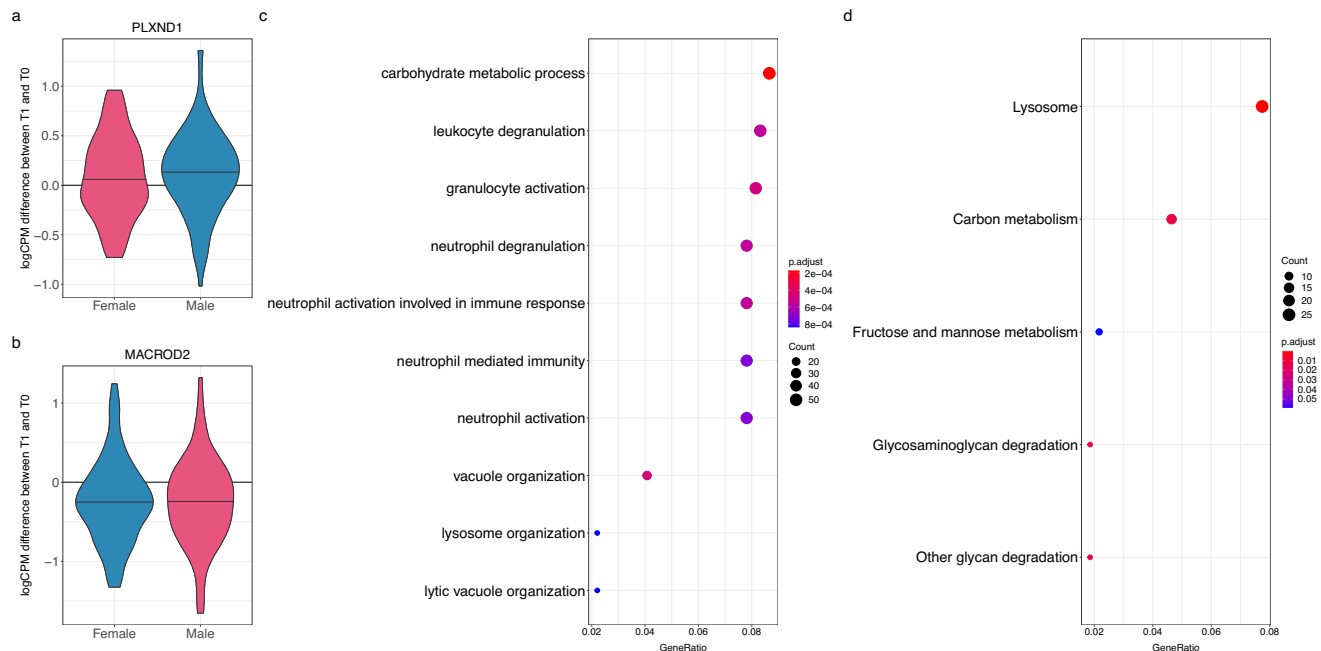

**Fig. 3 | Biological processes changing with time in peri-pubertal males and females. a**, **b** Violin plots of the distribution of normalized difference in gene expression between the two timepoints for each individual, plotted separately for each sex, **c**, **d** - dotplots represent enrichment of Gene Ontology biological processes (**c**) or KEGG pathways (**d**) within genes whose expression is higher in older children. Over-representation analysis performed using Clusterprofiler with a hypergeometric test with Benjamini–Hochberg adjustment for multiple comparisons.

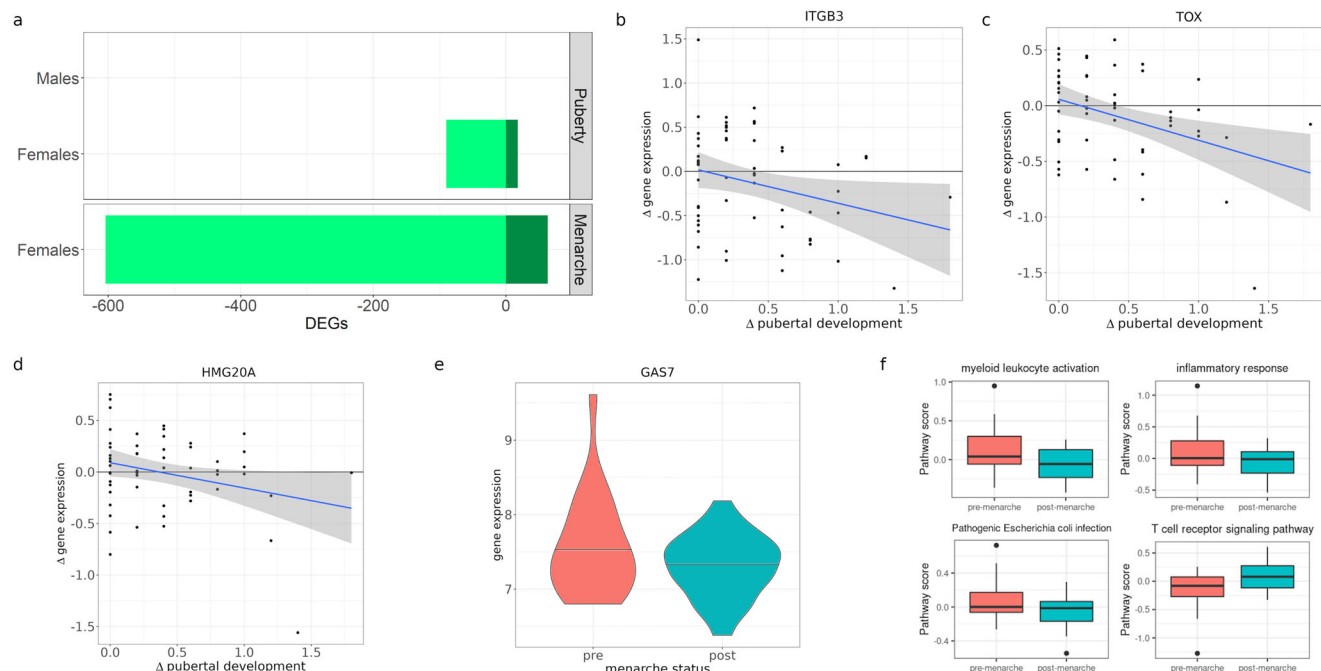

**Fig. 4 | Gene expression changes across puberty. a** Barplot with the number of genes differentially expressed (DEGs) with puberty and menarche (10% FDR). Light green denotes downregulated genes, dark green denotes upregulated genes. **b**–**d** Scatterplots represent longitudinal change in pubertal development (x axis) against log2-fold change in gene expression (y axis). Blue line represents best linear fit, and shaded grey area its 95% confidence interval. Rho denotes Spearman correlation and its *p*-value, **e** Violin plot represents log2CPM of GAS7 gene expression in pre- and post-menarcheal females. **f** Gene expression across biological processes and pathways differentially expressed between pre-menarcheal (*n* = 22) and post-menarcheal (*n* = 44) females. Center lines show the medians; box limits indicate the 25th and 75th percentiles; whiskers extend 1.5 times the interquartile range from the 25th and 75th percentiles; points outside the wiskers are represented by dots.

as females advance through puberty had also been found to be differentially methylated between pre- and post-pubertal stages. This overlap is not higher than expected by chance (Fisher's test *p*-value = 0.75). Of these, *ITGB3* was downregulated in children in later pubertal stages and also hypermethylated, while two others, *TOX* and *HMG20A*,

were downregulated but hypomethylated in children that were in a later pubertal stage (Fig. 4b–d). Genes upregulated as females advanced through puberty were enriched for biological processes related to negative regulation of protein modification (q-value=0.04). Additionally, several genes were involved in processes important for

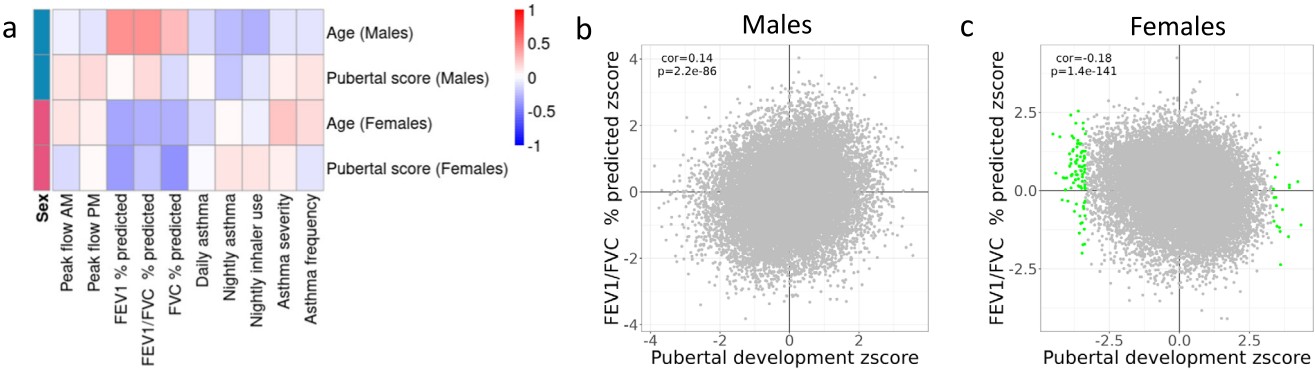

**Fig. 5 | Comparison of longitudinal effects of asthma and puberty on gene expression. a** Heatmap depicts pair-wise Pearson's correlations between normalized effects on gene expression (z-scores) of measures of asthma (rows) and puberty (columns). **b, c** Scatterplots of normalized effect of puberty (x axis) and FEV1/FVC percent predicted (y axis) on gene expression in males (**b**) and females (**c**). Inset values reflect Pearson's correlation coefficient and its two-sided *p*-value.

pubertal development in females. For example, *MGAT1* and *RPN1* are involved in N-glycan biosynthesis, reflecting previously-reported changes in serum concentration of N-glycans with age[44,45], *TSPO* enables cholesterol translocation into the mitochondria to allow steroid hormone synthesis, and *ACAA1* plays a role in fatty acid metabolism/degradation, potentially reflecting the changes in female body fat distribution that appear during puberty and characterize the adult body[46].

### Gene expression changes post-menarche

Menarche is an important milestone of pubertal transition marking sexual maturity in females. We utilized a cross-sectional study design to analyze whether gene expression differed between pre-menarche (females who have not yet experienced menarche at the time of the study, $N = 22$) and post-menarche females (females who are already menstruating at the time of the study, $N = 44$). A longitudinal analysis was not performed for menarche because this approach was only possible for 4 individuals in our cohort. We identified 667 differentially expressed genes, 604 genes had higher expression in pre-menarche females while 63 genes had higher expression in post-menarche females (Supplementary Data 11).

Genes downregulated after menarche were enriched for GO biological processes involved with immune responses, such as myeloid leukocyte activation ($q$-value $= 2.6 \times 10^{-6}$), and inflammatory response ($q$-value $= 4.3 \times 10^{-7}$) and KEGG pathways related to bacterial infection (e.g. Pathogenic Escherichia coli infection, Legionellosis $q$-value $= 0.05$). Genes with higher expression after menarche were enriched for the KEGG term T cell receptor signaling ($q$-value $= 0.08$), suggesting a shift from predominantly innate to adaptive immunity as females sexually matured (Fig. 4f, Fig. S3). Nine of the genes that changed expression past menarche had also been found to have methylation changes between pre- and post-puberty in females (LASP1, GAS7, CTSA, MYL9, AGO4, C1RL, GALNT2, TRAPPC1, C17orf62)[24]. This overlap is not higher than expected by chance (Fisher's test $p$-value $= 1$). All of these were downregulated post menarche, and six were hypermethylated in females post puberty (e.g., Fig. 4e). Genes changing methylation status between pre- and post-puberty had also been found to be enriched for genes harboring nearby estrogen response elements[24]. Estrogen and androgen are the main hormones driving the developmental changes during puberty in females and males, respectively. These hormones bind to their specific hormone receptors, which in turn bind the DNA at response elements and regulate the expression of target genes. The androgen receptor was not expressed in leukocytes in our cohort, however, the estrogen receptor gene (*ESR1*) was expressed both in males and females. To investigate whether estrogen may directly regulate gene expression in immune cells during puberty, we performed a motif scan for estrogen receptor

response elements in the cis-regulatory region of differentially expressed genes. We identified binding sites for the estrogen receptor in the regulatory region of 30 genes with transcriptional changes during puberty and 236 genes that were differentially expressed between pre- and post-menarcheal females. However, we observed no enrichment for genes with nearby estrogen receptor binding sites.

### Childhood asthma traits and gene expression changes during puberty

Asthma and other diseases with an immunological component change severity and disease manifestation during adolescence[13,14,18,47]. We used quantitative measures of asthma symptoms and severity to investigate whether the changes in gene expression in immune cells during puberty are also associated with changes in the disease status in our cohort. We considered longitudinal changes in asthma symptoms and investigated whether they are associated with longitudinal changes in gene expression. We found that for 83 genes expression is associated with asthma severity across male individuals. When considering overall correlation in gene expression changes that are associated both with puberty and asthma, we observed sex-specific patterns (Fig. 5a). In males, gene expression changes associated with puberty (and age) are inversely correlated with those associated with asthma symptoms and positively correlated with those associated with pulmonary function. Note that children with asthma generally have decreased pulmonary function, while significant correlations between effects of peak expiratory flow and age (and, likewise, pubertal development) are expected as peak expiratory flow rate increases with age and height[48]. For example, pulmonary function (e.g. FEV1/FVC percent predicted values) is positively correlated with puberty and age, indicating that genes upregulated in older males are also upregulated during puberty in males that have high pulmonary function (Fig. 5b). Conversely, in females gene expression changes associated with puberty (and age) are positively correlated with those associated with asthma symptoms and inversely correlated with pulmonary function (Fig. 5c).

### Genetic variation interacts with puberty to affect gene expression

Timing of puberty is influenced by both environmental[49] and genetic[50,51] factors acting primarily through their influence on gene expression[51]. We sought to identify genetic variants whose effect on gene expression changed throughout puberty. For example, an allele may increase gene expression in early puberty but decrease it in late puberty (qualitative change in the genetic effect on gene expression during puberty). A quantitative change in the genetic effect on gene expression during puberty would be defined instead as an allele that is associated with increased gene expression but the magnitude of the increase varies in different puberty stages. This type of analysis is

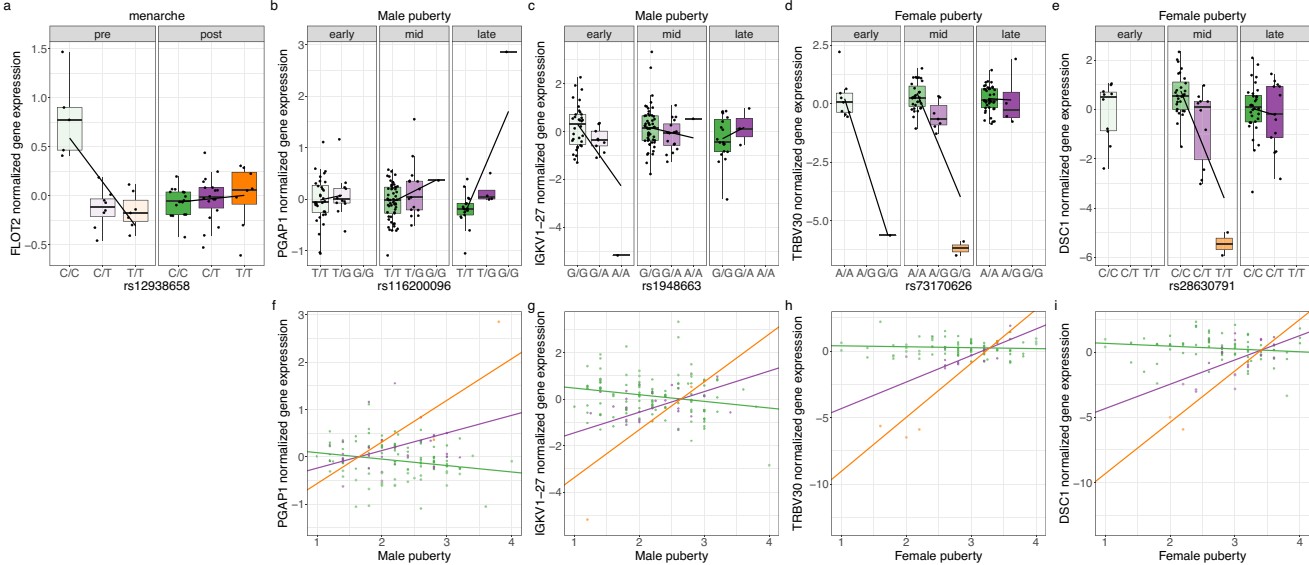

**Fig. 6 | Genetic variation interacts with pubertal status to influence gene expression.** Boxplots (top row) and scatterplots (bottom row) depict gene expression across the three genotype classes. **a** Menarche interacts with rs12938658 to affect the expression of the *FLOT2* gene, **b, f** Pubertal development stages interact with rs116200096 to affect the expression of the *PGAP1* gene in males, **c, g** Pubertal development stages interact with rs1948663 affect the

expression of the *IGKV1-27* gene in males, **d, h** Pubertal development stages interact with rs73170626 to affect the expression of the *TRBV30* gene in females, **e, i** Pubertal development stages interact with rs28630791 to affect the expression of the *DSC1* gene in females. Center lines of the box plots show the medians; box limits indicate the 25th and 75th percentiles; whiskers extend 1.5 times the inter-quartile range from the 25th and 75th percentiles.

also known as dynamic or context quantitative trait locus (QTL) mapping[52–67].

We first investigated whether genetic effects on gene expression changed across age. We did not find age to modify the effect of genetic variation on gene expression in either males or females. To find genetic variants whose effect on gene expression changes during puberty, we tested for interaction between puberty stage or menarche and genotype on expression of the nearby gene. We found that genetic effects on gene expression varied across puberty stages for four genes: *IGKV1-27* and *PGAP1* in males, and *TBV30* and *DSC1* in females. Additionally, in females, genetic effects on the expression of *FLOT2* varied pre- and post-menarche (Fig. 6, Supplementary Data 12). We observed the C allele at rs12938658 to increase the expression of the *FLOT2* gene in pre-menarcheal females, while its effect in post-menarcheal females trended in the opposite direction (Fig. 6a). *FLOT2* encodes a caveolae-associated integral membrane protein, and its expression has been implicated in asthma risk via Transcriptome-Wide Association Study (TWAS)[68].

We also found interactions between genotype and puberty on expression of B cell (*IGKV1-27* in males, Fig. 6b, f) and T cell (*TRBV30* in females, Fig. 6d, h) antigen-recognition proteins, with genetic effects on gene expression disappearing by late puberty. We had previously identified genotype-specific effects of lymphocyte and neutrophil fractions as well as nightly asthma symptoms on expression of *TRBV30* in the same cohort[69], which suggests that this association may be mediated by changes in blood cell composition that accompany puberty[70]. Interestingly, a sex-biased genetic effect on expression of *TRBV30* had been previously reported[71] with a significant effect in adult male but not female spleen. Our findings suggest that the genetic effect on *TRBV30* expression may be active in pre-pubertal females but disappear due to different mechanisms of gene expression regulation in post-pubertal females.

In males, we found an interaction between the pubertal development stage and the genotype at rs116200096 on the expression of the *PGAP1* (Fig. 6c, g) gene. This gene encodes an enzyme catalyzing the inositol deacylation of glycosylphosphatidylinositol, which ensures proper folding of proteins involved in embryogenesis,

neurogenesis, immunity, and fertilization[72]. Previously, we had found that expression of *PGAP1* was modulated by the interaction with genotype across many contexts including cell composition and psychosocial factors, such as the extent of self-disclosure of thoughts and feelings, and frequency of verbal arguments within the family[69]. Lastly, we observed a strong effect of the T allele at rs28630791 on the expression of the nearby *DSC1* gene only in mid-pubertal females (Fig. 6e and i). *DSC1* encodes desmocollin 1 - a cell surface protein forming part of the desmosome that ensures tight junctions between epithelial cells[73] but is also highly expressed by macrophages where it may impair HDL biogenesis[74].

## Changes in gene expression during puberty are associated with age at menarche and asthma risk

We have limited information on the genetic underpinnings of pubertal development. As a consequence, our knowledge of the molecular mechanisms accompanying developmental changes in the different body sites during puberty is limited. Our study identified changes in the expression of several genes in immune cells. To confirm that these gene expression changes are markers of pubertal development in independent cohorts, we considered GWAS data for age at menarche from the UK Biobank[51] jointly with gene expression data in whole blood from the GTEx study[75]. To investigate the expression of which genes is associated with age at menarche, we performed a Transcriptome-wide association study (TWAS). TWAS allows the integration of gene expression data with GWAS data to identify plausible candidate genes driving trait variability in the population. We performed TWAS for age at menarche by incorporating whole blood eQTL data from GTEx(v8) with a GWAS of age at menarche from 370,000 individuals[51] using SPrediXcan. Among 7174 tested genes, we detected a total of 108 genes associated with age at menarche at a genome-wide significant level (Bonferroni corrected *p*-value < 0.05, Fig. 7a, Supplementary Data 13). These genes associated with age at menarche were mostly enriched in immune-related pathways or diseases: MHC protein complex and autoimmune thyroid disease or antigen processing and presentation, (10% FDR). Notably, we found that 27 of the genes associated with age at menarche in the TWAS were undergoing

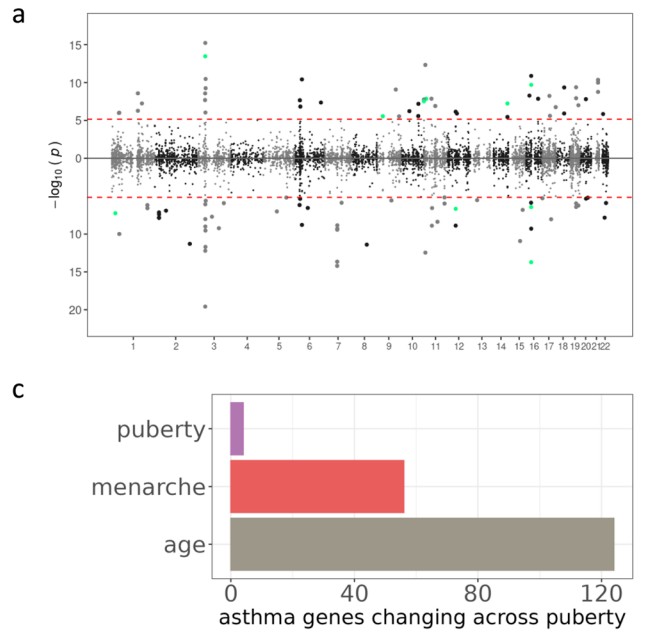

**Fig. 7 | Changes in gene expression during puberty are associated with age at menarche and asthma risk. a** Miami plot representing our TWAS analysis p-values for age at menarche across the genome (see Methods). Each dot is a gene; red line represents the Bonferroni significance threshold. Genes with higher expression in females undergoing menarche later are plotted above the x axis, genes with higher expression in females undergoing menarche earlier are plotted below the x axis. Shade of gray and x-tick marks separate chromosomes. Green color represents

TWAS genes which were also significantly downregulated in postmenarcheal females. **b** Expression of genes associated with age at menarche differs across puberty in females. Traits (left column) are connected to their respective differentially expressed genes (middle column) which are connected to the TWAS trait of age at menarche. Color represents the trait: age (grey), puberty (lilac) and menarche (red). **c** Barplot of the number of genes previously implicated in asthma whose expression levels differed across pubertal traits (y axis).

significant expression changes in peri-pubertal females: 17 genes with expression changes associated with age, 10 genes whose expression differed between pre- and post-menarcheal females, and one gene whose expression was higher when females were at later pubertal stages (Fig. 7b, Supplementary Data 14). For example, the gene *EEFSEC* (Selenocysteine-Specific Elongation Factor) is upregulated as females become older. Previous studies implicated this gene in endometrial cancer[76] and preterm birth[77], and a meta-analysis found a protective effect of late menarche on endometrial cancer risk[78].

Pubertal transition is associated with a shift in the sex distribution of asthma prevalence - in childhood males have higher asthma prevalence, while in adulthood, women have higher asthma prevalence and morbidity/severity[13,14,79]. A previous TWAS[68] associated expression of 1621 genes with asthma risk. We investigated whether pubertal development affected the expression of these genes. In both sexes, we observed gene expression changes over time during the peri-pubertal period for 124 genes associated with asthma risk. In addition, the expression of 56 genes associated with asthma risk differed between pre- and post-menarcheal females, and four genes associated with asthma risk changed expression as females progressed through puberty (Fig. 7c, Supplementary Data 15).

## Discussion

Previous genomic studies have focused on exploring DNA methylation changes between pre- and post-puberty[24–28] in peripheral blood immune cells. Yet, despite the clear evidence of epigenetic reprogramming during pubertal development, gene expression changes throughout puberty remain understudied. Here, we presented a comprehensive analysis of immune cell gene expression during the peri-pubertal period by analyzing the association of age, pubertal development stage, and menarche status with gene expression in a sample of 251 children with asthma. Unlike previous approaches, we sampled the entire span of pubertal development, thus capturing both early and late changes associated with puberty. Furthermore, we

augmented our longitudinal study design with cross-sectional analyses to corroborate findings on gene expression changes accompanying the peri-pubertal period.

Changes to human physiology across the lifespan have been extensively studied for decades[80–85]. Yet, the effect of age on gene expression in humans has not been explored thoroughly. Particularly, we lack insight into the transcriptional changes across the peri-pubertal period. Here, we discovered the widespread effects of age on immune cell gene expression in both males and females. To directly compare gene expression changes associated with age between the two sexes, we applied multivariate adaptive shrinkage, which revealed a remarkable consistency of age effect on gene expression between peri-pubertal males and females. Genes overexpressed when children were older were enriched for genes involved in biological processes related to viral gene expression and translation initiation, which may reflect the higher demand for protein anabolism when children are growing most rapidly[40]. We observed a small but significant enrichment of genes differentially expressed with age in our peri-pubertal sample within genes differentially expressed with age in the elderly[39], which implies that some age effects on gene expression may be universal across the lifespan, or at least shared between puberty and senescence. However, age had the opposite effect on gene expression in youth and the elderly for 35% of those genes, suggesting that senescence may be associated with reversal of some of the same physiological processes leading to sexual maturity.

While the epigenetic changes accompanying pubertal development have been explored previously in several studies[24–28], the likely subsequent transcriptional changes are largely unknown. We explored the gene expression changes in males and females tracked longitudinally through puberty. While we find a larger number of differentially expressed genes with age in males compared with females, we only detected significant longitudinal gene expression changes in females but not males across puberty stages. In our sample, females and males were of similar age. However, because females

experience puberty earlier than males, females in our sample were overrepresented at later pubertal stages while males were over-represented at earlier pubertal stages (Fig. 1c, t-test *p*-value = 3.54 × $10^{-6}$). Our results thus suggest that in immune cells, gene expression changes during later pubertal stages are more pronounced than those occurring during earlier pubertal development. Our sample size should be adequate to detect large changes at any stage if those were to occur. Thus despite puberty being an important time for developmental changes in both sexes, expression changes associated with pubertal development in males during the time frame considered in our dataset were relatively smaller when compared to those of females. Furthermore, our results are in line with previous findings of more genes differentially methylated between pre- and post-pubertal females than males[24,26,30]. Accordingly, our study adds to the existing evidence of a stronger effect of pubertal development on the functioning of the immune system in females compared to males. However, we found only a small overlap between genes changing expression with pubertal development in females and those previously reported to change methylation status between pre- and post-puberty in females[24]. One biological explanation for this lack of enrichment may be that some genes are epigenetically poised for transcription even before puberty but are not expressed until activated by sex hormones. Alternatively, the small overlap between genes differentially expressed as females advance through puberty and genes associated with regions differentially methylated in pre- and post-pubertal females may suggest that some of the differentially expressed genes we discovered are specific to the pubertal transition period.

Expectedly, gene expression changes associated with pubertal development in females were well correlated with age effects. This finding helps further validate the use of self-reported pubertal development scores in research settings. However, we discovered more genes differentially expressed with age than pubertal development. This could be because some of the females in our sample had completed pubertal development and only differed by age, thus allowing us to discover additional genes for which age effects may be subtler but continue past puberty.

A recent study of DNA methylation changes associated with the five markers of pubertal development used in our study found that menarche status was the puberty marker associated with the highest number of regions differentially methylated[26]. In agreement with this finding, we found more genes whose expression differed between pre- and post-menarcheal females compared to genes discovered to change expression as females progress through pubertal development. This is likely because the onset of first menstruation is a less ambiguous marker of the shift into sexual maturity.

Previous studies of DNA methylation changes between pre- and post-puberty in females found an overrepresentation of genes involved in immune and inflammatory responses[24,25,27], reproductive hormone signaling[24], and other physiological changes related to growth and development[26]. Our results allowed a more fine-grained insight into the changes in the immune system that accompany the sexual maturation of females. We discovered an enrichment of processes related to innate immunity within genes more highly expressed in pre-menarcheal females and enrichment for T cell receptor signaling in genes more highly expressed in post-menarcheal females. The latter result corroborates a previous finding of T cell receptor signaling being overrepresented within genes near differentially methylated regions, which were co-expressed in post-menarcheal females[24]. Our results suggest a shift from innate to adaptive immunity past menarche. This is supported by the known effects of sex hormones on the immune system. For example, estrogen has been shown to promote adaptive immunity[86] and may promote or restrict innate immunity[87].

Previous cross-sectional studies of asthma severity and age in large cohorts have shown that prior to puberty, the incidence of

asthma is higher in males, and then post-puberty, adults with asthma are more likely to be female[14]. However, a recent longitudinal study throughout adolescence found that asthma severity decreased equally in male and female subjects, probably due to an increase in androgens in both sexes[47]. Androgen levels were found to be positively associated with increased pulmonary function[18,88], as well as suppression of airway inflammation[89,90]. Due to the demographics of our cohort, we captured late pubertal stages in females, which are associated with increased incidence of asthma[14], thus resulting in the sex-specific patterns of gene expression that we observed. Indeed, we found that gene expression changes in late puberty in males are positively correlated with gene expression changes associated with pulmonary function and inversely correlated with changes associated with asthma symptoms, while in females, these patterns are reversed. Furthermore, the genes expressed in late pubertal stages in females are enriched for adaptive immunity processes, which have been found to be influenced by estrogen[86]. This is a potential mechanism for previously observed associations between estradiol and decreased pulmonary function in adolescent females with asthma[18] and could be further investigated in conjunction with hormone level measurements in future studies.

While sex-biased genetic effect on gene expression had been reported before[71], previous studies included only adult participants. Here we demonstrate that genetic regulation of gene expression varies during pubertal development. For example, the genetic effect on the expression of the B cell antigen recognizing protein *IGKV1-27* gene disappears by late puberty in males. We found hormone response elements for androgen and estrogen receptors near *IGKV1-27*, suggesting that the different hormonal context between early and late puberty may explain the observed interactions between genetic effect and puberty at this locus. In females, we discovered that the genotype at rs73170626 is associated with the expression of *TRBV30* gene in early puberty, but this genetic effect disappears in late puberty. This gene encodes a T cell receptor. Interestingly, a sex-biased eQTL was previously reported for this gene, with genetic effects on gene expression present only in adult males but not females[71]. This implicates puberty in origins of sex-biased genetic effects on gene expression, which may translate into disease risk in women later in life.

A previous TWAS identified 205 genes whose expression in neural tissues is associated with age at menarche[51]. Using a similar TWAS approach here we demonstrated an association between blood expression levels of 108 genes and age at menarche. For 31 of these genes, our results confirmed previous observations for causal or pleiotropic effects between blood gene expression and age at menarche that were based on Mendelian randomization. This overlap represents a significant enrichment over chance (OR = 195, *p*-value < 2.2e-16)[51]. We demonstrated that expression of 27 genes associated with age at menarche changed across puberty, including a gene implicated in endometrial cancer risk[76], which may help explain the link between earlier age at menarche and increased endometrial cancer risk[78]. Additionally, we demonstrated that pubertal transition was associated with changes in expression of genes associated with asthma risk. Further investigations of genes that are both associated with asthma risk and differentially expressed between pre- and post-menarche females may provide clues to understanding increased asthma risk in females post puberty.

Our study is the first genome-wide characterization of patterns of gene expression changes in peri-pubertal males and females, and complements the existing body of literature on the epigenetic reprogramming of immune cells during puberty. Our results also add to the literature on lifespan changes in gene expression by focusing on early adolescence. Overall, our findings suggest a shift from a more innate to a more adaptive immune response past puberty in females, which might be a mechanism for the higher incidence of autoimmune disease in adult women compared to men. Additionally, we demonstrate that the associations between gene expression and pubertal development

can be modified by genetic variation, which may contribute to inter-individual variability in incidence and severity of diseases with peri-pubertal onset.

## Methods

### Study participants
Participants were drawn from the Asthma in the Lives of Families Today study (ALOFT; samples collected November 2010-July 2018). The study was approved by Wayne State University Institutional Review Board, decision #0412110B3F. This multi-ethnic cohort was established to investigate the effects of psychosocial experiences on asthma symptoms in youth living in the Detroit Metropolitan Area. To be included in the study, youth were required to be between 10 and 17 years of age at the time of recruitment and diagnosed with at least mild to persistent asthma by a physician (with diagnosis confirmed from medical records). The total sample included in the current study consisted of 251 participants (103 females and 148 males) aged 10-17 at recruitment for whom we successfully collected RNA-seq and genome-wide genotype data (cohort described previously[69]). Written assent and consent were obtained from the participating youth and their parent, respectively. Participants were compensated. The ALOFT study included three measurements at annual intervals, but only the first two sampled time points were included for individuals with more than two longitudinal data points in the ALOFT study. This strategy allowed us to maximize the sample size for the longitudinal analysis. For 65% of the participants, we were able to collect longitudinal gene expression data from at least two timepoints. The average time interval between the two-time points was 1.27 years (SD = 0.46 years).

### Gene expression data
Leukocyte gene expression data for all timepoints and genotypes imputed to the whole genome were obtained as described previously[69]. In brief, RNA was extracted using LeukoLOCK (Thermo Fisher) and only samples with RNA Integrity Number (RIN) of at least 6 measured on Agilent Bioanalyzer were included. Library preparation was done in batches of up to 96 samples following standard Illumina TruSeq Stranded mRNA protocol, with longitudinal samples from the same individual processed together in the same batch. Samples were sequenced on the Illumina NextSeq500 Desktop Sequencer to obtain 75 bp paired-end reads. A total of 21 million (M) to 76 M reads were collected per sample, with a mean of 41 M reads/sample. The RNA-seq data were aligned to the human genome version GRCh37 using HISAT2[91] and counted across all genes using HTSeq. Samples with excess PCR duplicate rate (>60%) were excluded and sample identity was confirmed by comparing the genotypes obtained from the RNA-seq data with those obtained from the DNA samples. For all gene expression analyses, genes on sex chromosomes and genes with expression below 6 reads or 0.1 counts per million in at least 20% of samples were dropped. Data for the first time point for each individual and longitudinal data for 120 of the participants have been previously published[69], and together with additional new data included in this study are accessible at dbGaP accession #phs002182.v2.p1.

### Genotype data
VCF files were generated from low-coverage (~0.4×) whole-genome sequencing of one blood-derived DNA sample per individual and imputed to 37.5 M variants using the 1000 Genomes database by Gencove (New York, NY). These data are accessible at dbGaP accession #phs002182.v2.p1.

### Puberty measurements
Puberty staging was self-reported as this has been shown to be reliable, less intrusive, and allows for large epidemiological studies[92,93]. Pubertal development was assessed at each timepoint prior to the blood draw for gene expression measurement via a sex-specific self-report questionnaire[31]. The questions and possible responses can be found in Table S2. The overall pubertal development score is a mean of the responses to specific questions 1–5, given that at least four valid responses are provided. Pubertal development stages are analogous to the Tanner stages[42,43] and were treated as a continuous variable in all analyses. For female participants, we also considered self-reported menarche status (pre-menarche/post-menarche).

### Asthma measurements
Measures of asthma symptoms and severity were collected at each timepoint prior to the blood collection as described previously[69]. Briefly, we used measures of lung function collected via spirometry (FEV1 percent predicted, FVC percent predicted, FEV1/FVC percent predicted) and peak expiratory flow meter (morning and evening peak expiratory flow averaged over four days), and self-reported measures of asthma symptoms, severity and inhaler use.

### Differential gene expression analysis
**Longitudinal analysis.** We used DESeq2 to test for gene expression changes over time, across puberty stages and with asthma traits in the longitudinal sample. Specifically, we used the following models:

gene expression ~ fraction of reads mapping to exons + fraction of high-quality reads + RIN + participant ID + age

gene expression ~ fraction of reads mapping to exons + fraction of high-quality reads + RIN + participant ID + pubertal stage

gene expression ~ fraction of reads mapping to exons + fraction of high-quality reads + RIN + participant ID + age + asthma measure

In the DESeq model, we corrected for fraction of reads mapping to exons, fraction of high-quality reads (after removing PCR duplicates), RNA Integrity Number (RIN), and individual effect.

These analyses were run in each sex separately. A subset of 24 participants reported a lower pubertal development score in the second timepoint compared to baseline. To account for these mis-reported data, we corrected the second timepoint to match the pubertal development score measured for the same individual at baseline.

Significance threshold for differentially expressed genes was set at 10% FDR using the Benjamini–Hochberg correction method.

**Cross-sectional analysis.** We used DESeq2 to test for gene expression associated with sex, age, puberty, and menarche status in each sex separately. To account for possible confounding factors, we corrected for library preparation batch, fraction of reads mapping to exons, fraction of high-quality reads, RNA Integrity Number (RIN), and the top three genotype PCs.

gene expression ~ batch + fraction of reads mapping to exons + fraction of high-quality reads + RIN + genotype PCs + **age**

gene expression ~ batch + fraction of reads mapping to exons + fraction of high-quality reads + RIN + genotype PCs + **pubertal stage**

gene expression ~ batch + fraction of reads mapping to exons + fraction of high-quality reads + RIN + genotype PCs + **menarche**

**Multivariate adaptive shrinkage (mash).** To investigate gene expression changes associated with age in both sexes, we used the multivariate adaptive shrinkage (mash) method[33] using the mashr package v.0.2.40 in R v4.0. As input, we provided the log-fold change and corresponding standard error from the DESeq analysis in each sex, subset for genes tested in both sexes (18341 genes). We considered genes with local false sign rate (LFSR) < 0.1 to be differentially expressed.

### Hormone receptor binding site analysis
To test for enrichment of estrogen receptor binding sites near differentially expressed genes, we downloaded position weight matrices (PWM) for ESR1 from the JASPAR Vertebrate Core Motifs database version 2020[94]. We counted the number of genes with at least one

strongly-predicted (log(2) odds over genomic background>10) binding motif within 10 kb of the transcription start site within the set of differentially expressed genes and the set of all tested genes for each binding motif separately. We tested for enrichment using Fisher's exact test and used *P*-value < 0.05 as threshold for significance.

## Gene ontology enrichment analysis

We used the clusterProfiler[95] package for R to run Gene Ontology enrichment analysis for the differentially expressed genes for each variable. We used as background all genes tested for differential expression for that same variable The significance threshold was set at 10% FDR, and we only considered categories that included at least three genes from the test set.

To investigate pathway-level changes between pre- and post-menarcheal females, we calculated pathway scores of interest as follows: (1) calculated residual expression values by regressing the effects of confounders used in differential gene expression analysis; (2) extracted the subset of differentially expressed genes between pre- and post-menarcheal females and belonging to a specific GO term or KEGG pathway; (3) scaled mean-centered gene expression for each gene across individuals; (4) calculated the average values as a specific pathway score across genes for each individual.

## Interaction eQTL mapping

To prepare the data for eQTL mapping, we quantile-normalized the gene expression data for the cross-sectional sample of 251 participants using the voom(method = "quantile") function in *limma* (Law et al. 2014). To remove the effect of technical confounders, we regressed the effects of: RIN, fraction of reads mapping to exons, fraction of high-quality reads, data collection batch, library preparation batch, genotype PC1, genotype PC2, genotype PC3, age, weight, and height. We first performed eQTL mapping in both sexes combined to identify genetic variants associated with gene expression levels. To do this, we used FastQTL[96] with adaptive permutations (1000-10,000). For each gene, we tested all genetic variants within 1 Mb of the transcription start site (TSS) and with cohort MAF > 0.1. We defined significant eGenes atFDR<10%. For interaction eQTL mapping, we tested the SNP with the lowest p-value for each eGene. We quantile-normalized the gene expression data in each sex separately using the voom(method = "quantile") function in *limma* and regressed the effects of: RIN, fraction of reads mapping to exons, fraction of high quality reads, library preparation batch, genotype PC1, genotype PC2, genotype PC3. To identify genetic variants that interacted with puberty stage, age or menarche status to influence expression of the nearby gene, we tested the interaction between genotype dosage and each of these variables while controlling for the marginal effects using the following models in each sex separately:

Normalized gene expression ~ genotype dosage + age + **genotype dosage × age**

Normalized gene expression ~ genotype dosage + pubertal stage + **genotype dosage × pubertal stage**

Normalized gene expression ~ genotype dosage + menarche + **genotype dosage × menarche**

We used 1000 permutations to correct the *p*-values as described previously[69] before applying Storey's *q*-value method to control for FDR.

## Transcriptome-wide association study of age at menarche

We performed TWAS for age at menarche (AAM) by running the module SPrediXcan in the MetaXcan software v0.6.11[97]. To perform TWAS, we first downloaded GWAS summary statistics of AAM from the ReproGen consortium (https://www.reprogen.org/)[51], which were generated based on data from 329,345 women from 40 studies. This data set contains over 10 million genetic markers. After transferring marker name from "chr:position" to "rs id" using annovar pipeline[98],

over 4 million markers remained in the following TWAS analysis. As input for SPrediXcan, we also provided covariances file (single-tissue LD reference files) and gene expression model in the blood tissue from the GTEx (V8) project[75], available from (https://predictdb.org/). We used gene expression models trained in the blood tissue based on the Elastic Net model from the GTEx (V8) project. To validate our results, we used Fisher's exact test to consider the enrichment of genes associated with age at menarche through TWAS within genes previously implicated in age at menarche through Mendelian randomization[51].

## Reporting summary

Further information on research design is available in the Nature Portfolio Reporting Summary linked to this article.

## Data availability

The phenotype, genotype and gene expression data (RNA-seq fastq files) used in this study are available on dbGAP under accession number: phs002182.v2.p1. All other data generated or analyzed during this study are included in this published article (and its supplementary information files) and source data are provided as a Source Data file. Source data are provided with this paper.

## Code availability

The scripts detailing the analyses conducted in R are available at https://github.com/piquelab/ALOFT_puberty.

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

## Acknowledgements

We thank Carole Ober for helpful comments on an earlier version of this manuscript, and members of the Luca, Pique-Regi and Zilioli groups for helpful discussions. This study was supported by National Heart, Lung, and Blood Institute grants RO1HL114097 to R.S. and S.Z., RO1HL162574 to F.L., R.P.R., and S.Z. The graphical representation of the male and female body during puberty used in Fig. 1 were created with BioRender.com.

## Author contributions

Conceptualization and Funding acquisition: R.B.S., R.P.R., F.L., and S.Z.; Resources: X.W., R.B.S., S.Z., R.P.R., and F.L.; Methodology: X.W., R.B.S., R.P.R., and F.L.; Project administration: R.P.R. and F.L.; Data curation: J.R., J.B., R.H., A.A., and H.M.M.; Formal analysis, Investigation and

Visualization: J.R., J.C., and J.W.; Writing - original draft: J.R., J.C., R.P.R., and F.L.; Writing - review and editing: J.R., J.C., S.N., J.W., X.W., R.B.S., S.Z., R.P.R., and F.L. J.R. and J.C. contributed equally to this work.

## Competing interests

We declare that the authors have no competing interests as defined by Nature Research, or other interests that might be perceived to influence the interpretation of the article.
