## [Peer Review File · Nature Communications]

Analysis of transcriptional changes in the immune system associated with pubertal development in a longitudinal cohort of children with asthmaREVIEWER COMMENTS

Reviewer #1 (Remarks to the Author):

Resztrak et al. studied a large cohort of children ages 10-17 to identify transcriptional changes in leukocytes during aging, puberty and (in females) pre and post menarche. They compare the differentially expressed genes to previously published lists of genes changing between people at their 80s and 90s, and regions changing their methylation during puberty.

This is a useful and rich resource, some of which was previously published, but not analyzed with regards to those factors.

There is one concern that is not addressed in the paper and is almost hidden - the fact that all participants have asthma. This should be stated in the abstract and methods, and any potential effect on the findings should be discussed.

Other than that, several points to be considered and/or fixed, all minor

- "The difference in the number of differentially expressed genes in each sex likely reflected higher statistical power in boys than girls" – this claim can be tested directly by repeatedly down-sampling the boys to the same number and counting the number of identified DEGs.
- The correlation of fold change between sexes and the shared upregulated genes in both sexes seem to be missing. Consider adding.
- Boys&girls and males&females are used alternatively – stick to one.
- Fig2a – how is the age on the left of the figure related?
- Fig 2c – axis legend font too small. Add boys-girls correlation
- Fig S2 – axis legend font too small. Yaxis $-\log_{10}$ of p-value, not p-value.
- P5 – "Genes differentially expressed in our longitudinal peri-pubertal sample were enriched within genes differentially expressed in the elderly" – how many DEGs were identified in the elderly? What is the size of the overlap? Phrase the numbers more explicitly – how do the 76 and 44 relate to the 126?
- P6 – "Three of the 108 genes changing expression as girls advance through puberty had also been found to be differentially methylated between pre- and post-pubertal stages." - Is this size of overlap (3 out of 108 and 338) bigger than expected by random?
- Supplementary files containing tables should be in Excel, and each column should be labeled and if necessary, explained. In the one I looked for (Supp file S11), only gene ENSEMBL IDs were shown – it will make those files much more useful if gene symbols will be added. In S14 and s15, consider adding SNPs ids and effect size, etc. Will also be nice to call each file by its name and add a title, to help the reader that is downloading all those files know what they are.
- Add the P-value whenever an enrichment of a gene list to a GO term is stated.
- P7 – "Genes which differed between pre- and post-menarcheal girls were also more likely to be differentially expressed between boys and girls" – those upregulated or downregulated? In the same direction? I would expect boys and pre-menarcheal girls will be more similar, but needs to be shown. Were those genes "differentially expressed between boys and girls" defined and listed before?
- P7 – "Nine of the genes that changed expression past menarche had also been found to have methylation changes between pre- and post-puberty in girls²². All of these were downregulated past menarche, and six were hypermethylated in girls post puberty (e.g., Fig. 4E)." same as above regarding expected overlap size, and also gene names need to be specified somewhere, not only the one gene in fig 4e.
- P8 – consider replacing environment with a more appropriate term. Maybe phenotypes?
- Supplementary file 13 not found - http://genome.grid.wayne.edu/puberty/Asthma-TWAS_Blood.csv
- Figure 1C – score range is 0-4, consider matching x tick labels.
- The discussion discusses the reason for larger difference in girls, but the results in fig 2a and e show more DEGs in males, or a comparable number, respectively. Please clarify.
- "longitudinal samples from the same individual processed together in the same batch" – this also means that the first time point was always stored for a ~year more. Can the author comment on an expected effect, or show there is not effect of storage time on samples otherwise similar?
- Was there any filtering of the genes done prior to modeling? If so please specify.
- How were "all expressed genes in each sex" defined? What is their number?
- Figure 6 – how was association with asthma risk estimated? Are you sure color in A represents what is stated in the legend and not chromosome number? What do the colors in B stand for?

Reviewer #2 (Remarks to the Author):

Dr Resztak and colleagues have studied changes in gene expression in key processes including immune regulation during pubertal development. The manuscript is well considered and fluently written, and provides some important new insights into the changes in gene expression during pubertal development. I do have some concerns about study design which are detailed below. Major comments:

The study was conducted in a cohort of patients with asthma, which was presumably a pragmatic decision rather than an intended exploration of gene expression in pubertal development in children with asthma. It remained unclear to me how much the intention of the study was to explore changes in immune system gene expression and how much this was a more global exploration of key genes that are up and downregulated during this developmental period. What impact do the authors feel the particular cohort having asthma might have had on the study outcomes, particularly the assessment of immune system gene expression changes with genetic variation?

The definition of the pre- and post groups in which the comparisons were undertaken were surprising. To include girls that have had menarche in the pre group seems counter-intuitive for a study aimed to detect difference pre and post puberty. The explanations given on which groups were analysed for which experiments were not sufficient – for the 'Genetic variation interacts with puberty to affect gene expression' section testing was clearly on puberty stage or menarche; but for the preceding experiments the associations seem much more likely to be due to age than to pubertal stage given that there were late pubertal children included in the first timepoint group.

Minor comments:

Please provide line and page numbering in submitted documents in order to assist reviewing.

Title – should make reference to the immune system

Abstract – 'change in a sign of' is not very clear, please rephrase

Introduction 'the average age range for puberty...' this is not very accurate, are we speaking here about age range for onset, mid point of puberty or completion? As per reference

<https://doi.org/10.1210/er.2018-00248> 'In most populations 95 percent of girls experience onset of pubertal development between 8.5 and 13 years and the same percentage of boys between 9 and 13.5 years'

Results – Figure 1 – the 'early puberty' and 'late puberty' labelling of panel A might be confusing and be misinterpreted as 'children with early ie precocious or late ie delayed puberty'. Perhaps change to 'Early puberty stage' or similar

Figure 1 D – I am surprised that any of the 'pre' group have menarche as this is a late event in puberty and should place these girls in the late stage of menarche group. Onset of menarche should be used as an exclusion criterion for being in the 'pre' group, otherwise comparisons will not reflect pubertal changes but only those due to age or other factors

Results – 'Mutations near this gene have been implicated in a form of congenital hypogonadism 33–35' - close chromosomal proximity does not infer an aetiological connection here.

Discussion – 'Thus an alternative explanation would be that the gene expression changes associated

with pubertal development in boys were very subtle compared to those of girls during the time frame analyzed.' This seems unlikely given this is a period of major developmental change for both genders.

Reviewer #3 (Remarks to the Author):

The study a longitudinal assessment of gene expression by gender over time and is an interesting, novel analysis where they identified genetic effects on gene expression that change during pubertal development and differend by gender and in females with menarche a change in the asthma-associated gene- which may explain changes in females that may increase asthma after puberty and provide mechanistic insights.

Their methods are interesting using a multi-variate adaptive shrinkage approach that uses correlations by gender to improve effect size estimates and power. They also found significant genes associated with age and thymocyte development

Major comments:

1) Overall, the study is interesting, but it is so jam packed with information and it would be helpful to have a clear theme and focus. If they are trying to understand expression related to asthma and how it relates to asthma, that should be listed upfront as main objective. Is there any data on the cohorts as to whether the differentially expressed genes with asthma actually correlated with health outcomes? This would help bring context to the findings.

2) I would definitely have a bioinformatic/statistician review the methods. It appears there are so many comparisons, it would be difficult to know if the approach is solid.

3) Clarity on the meanings of the interactions between genotype and puberty should be described.

4) There are a couple of references missing that should be discussed and described in the context of how these related to clinical findings as relates to age and pubertal changes in asthma.

Severe asthma during childhood and adolescence: A longitudinal study.

Ross KR, Gupta R, DeBoer MD, Zein J, Phillips BR, Mauger DT, Li C, Myers RE, Phipatanakul W, Fitzpatrick AM, Ly NP, Bacharier LB, Jackson DJ, Celedón JC, Larkin A, Israel E, Levy B, Fahy JV, Castro M, Bleecker ER, Meyers D, Moore WC, Wenzel SE, Jarjour NN, Erzurum SC, Teague WG, Gaston B.J Allergy Clin Immunol. 2020 Jan;145(1):140-146.e9. doi: 10.1016/j.jaci.2019.09.030. Epub 2019 Oct 14. PMID: 31622688 Clinical Trial.

Effects of endogenous sex hormones on lung function and symptom control in adolescents with asthma.

DeBoer MD, Phillips BR, Mauger DT, Zein J, Erzurum SC, Fitzpatrick AM, Gaston BM, Myers R, Ross KR, Chmiel J, Lee MJ, Fahy JV, Peters M, Ly NP, Wenzel SE, Fajt ML, Holguin F, Moore WC, Peters SP, Meyers D, Bleecker ER, Castro M, Coverstone AM, Bacharier LB, Jarjour NN, Sorkness RL, Ramratnam S, Irani AM, Israel E, Levy B, Phipatanakul W, Gaffin JM, Gerald Teague W. BMC Pulm Med. 2018 Apr 10;18(1):58. doi: 10.1186/s12890-018-0612-x. PMID: 29631584 Free PMC article.

Baseline Features of the Severe Asthma Research Program (SARP III) Cohort: Differences with Age.

Teague WG, Phillips BR, Fahy JV, Wenzel SE, Fitzpatrick AM, Moore WC, Hastie AT, Bleecker ER, Meyers DA, Peters SP, Castro M, Coverstone AM, Bacharier LB, Ly NP, Peters MC, Denlinger LC, Ramratnam S, Sorkness RL, Gaston BM, Erzurum SC, Comhair SAA, Myers RE, Zein J, DeBoer MD, Irani AM, Israel E, Levy B, Cardet JC, Phipatanakul W, Gaffin JM, Holguin F, Fajt ML, Aujla SJ, Mauger DT, Jarjour NN. J Allergy Clin Immunol Pract. 2018 Mar-Apr;6(2):545-554.e4. doi: 10.1016/j.jaip.2017.05.032. Epub 2017 Aug 31. PMID: 28866107 Free PMC article.

This could be modified depending on the focus and direction of the paper.

5) The discussion of reprogramming and changes and consideration of why it explains autoimmune disease seems that there are a few themes being described from the findings and it is confusing

6) if the entire cohort had asthma- it would be helpful to see if any of the findings held true in a non asthma cohort

7) how was asthma defined in the cohort. Many basic details are missing.

8) how was puberty assessed.. Were there trained clinicians on Tanner staging? Were hormonal levels ascertained as well?

Minor comments:

The abstract doesn't fit a normal structure and has examples that again detract from the paper. What is the overall purpose—It seems it is trying to analyze changes in puberty and come with all types of associations and relationship with diseases without a clearly driven hypothesis or approach of questions they want to answer.

We thank the reviewers for their insightful comments and suggestions which we believe have greatly improved the quality of our manuscript. Below are point by point responses (in blue) to the their comments.

Reviewer #1 (Remarks to the Author):

Resztak et al. studied a large cohort of children ages 10-17 to identify transcriptional changes in leukocytes during aging, puberty and (in females) pre and post menarche. They compare the differentially expressed genes to previously published lists of genes changing between people at their 80s and 90s, and regions changing their methylation during puberty.

This is a useful and rich resource, some of which was previously published, but not analyzed with regards to those factors.

There is one concern that is not addressed in the paper and is almost hidden - the fact that all participants have asthma. This should be stated in the abstract and methods, and any potential effect on the findings should be discussed.

We thank the reviewer for this suggestion. We have now added a new section to the introduction, additionally we have added new sections to results, methods and discussion where we directly compare gene expression changes associated with both asthma and puberty effects.

Introduction:

"Sex hormones are hypothesized to play a role in the switch of sex skew, as asthma symptoms are also known to worsen around menstruation^{19,20}. Recently, a Mendelian randomization study showed a protective effect of increased sex hormone-binding globulin (SHBG) levels on asthma onset, with a larger effect in females than males²¹. Furthermore, in a cross-sectional study of a large cohort of children with severe asthma, circulating levels of androgens were found to positively associate with improved lung function and symptom control in pubescent males, while late puberty in females was associated with increased estradiol levels and decreased lung function¹⁵ Despite sex differences in susceptibility to asthma before and after puberty onset¹³, earlier onset of puberty has been found to increase the risk of asthma in both sexes¹⁷. "

The title of the new results section is

"Childhood asthma traits and gene expression changes during puberty"

Discussion:

"Previous cross-sectional studies of asthma severity and age in large cohorts have shown that prior to puberty, the incidence of asthma is higher in males, and then post-puberty, adults with asthma are more likely to be female¹⁴. However, a recent longitudinal study throughout adolescence found that asthma severity decreased equally in male and female subjects, probably due to increase in androgens in both sexes⁷⁵. Androgen levels were found to be positively associated with increased pulmonary function^{15,76}, as well as suppression of airway inflammation^{77,78}. Due to the demographics of our cohort, we captured late pubertal stages in females, which are associated with increased incidence of asthma¹⁴, thus resulting in the sex-specific patterns of gene expression that we observed. Indeed, we found that gene expression changes in late puberty in males are positively correlated with gene expression changes associated with pulmonary function and inversely correlated with changes associated with asthma symptoms, while in females these patterns are reversed. Furthermore, the genes expressed in late pubertal stages in females are enriched for adaptive immunity processes which have been found to be influenced by estrogen⁷³. This is a potential mechanism for previously observed associations between estradiol and decreased pulmonary function in adolescent females with asthma¹⁵ and could be further investigated in conjunction with hormone level measurements."

Other than that, several points to be considered and/or fixed, all minor

- "The difference in the number of differentially expressed genes in each sex likely reflected higher statistical power in boys than girls" – this claim can be tested directly by repeatedly down-sampling the boys to the same number and counting the number of identified DEGs.

We thank the reviewer for this comment, yet down-sampling and ensuring that samples match by age and puberty stages is not immediately obvious. We rephrased this sentence to: "Genes differentially expressed as females grew older were enriched among genes differentially expressed in males (OR=9, p-value= 6.5×10^{-16} , Fig. 2B), suggesting a similar effect of age on gene expression in both sexes, despite the difference in the number of significant genes. "

- The correlation of fold change between sexes and the shared upregulated genes in both sexes seem to be missing. Consider adding.

We have added this information to the text:

"We found that these gene expression changes were largely shared between sexes, with 721 genes upregulated as children grew older and 1404 genes downregulated as children grew older in both sexes (Spearman rho=0.95, p-value $<2.2 \times 10^{-16}$, Fig. 2F, Fig. S3)"

- Boys&girls and males&females are used alternatively – stick to one.

Thank you for this comment. We decided to use females and males throughout the manuscript to reflect the language used in the self-reported questionnaire.

- Fig2a – how is the age on the left of the figure related?

We apologize for the oversight; it is actually superfluous and we have removed it.

- Fig 2c – axis legend font too small. Add boys-girls correlation

We have increased the font size, moved figure S3 as a part of 2c and added the correlation value between boys and girls to the new figure.

- Fig S2 – axis legend font too small. Yaxis -log10 of p-value, not p-value.

We have increased the font size and used -log10 of p-value which is standard for volcano plots. This has been fixed in the caption.

- P5 – "Genes differentially expressed in our longitudinal peri-pubertal sample were enriched within genes differentially expressed in the elderly" – how many DEGs were identified in the elderly? What is the size of the overlap? Phrase the numbers more explicitly – how do the 76 and 44 relate to the 126?

We have increased the clarity of these results as follows:

"Of the 1291 genes changing expression with age in the elderly, 188 were also differentially expressed in both sexes in our longitudinal peri-pubertal sample (OR = 1.2, p-value = 0.03). Among the genes differentially expressed in both youth and elderly, 44 genes increased expression and 76 decreased expression as a function of age in both groups."

- P6 – "Three of the 108 genes changing expression as girls advance through puberty had also been found to be differentially methylated between pre- and post-pubertal stages." - Is this size of overlap (3 out of 108 and 338) bigger than expected by random?

We found no enrichment between genes changing expression as girls advance through puberty and those previously found to be differentially methylated between pre- and post-pubertal stages. We have updated the text to reflect this finding:

“This overlap is not higher than expected by chance (Fisher’s test p-value=0.75).”

- Supplementary files containing tables should be in Excel, and each column should be labeled and if necessary, explained. In the one I looked for (Supp file S11), only gene ENSEMBL IDs were shown – it will make those files much more useful if gene symbols will be added. In S14 and S15, consider adding SNPs ids and effect size, etc. Will also be nice to call each file by its name and add a title, to help the reader that is downloading all those files know what they are.

We have followed the reviewers' suggestion and modified the supplementary tables as requested.

Table S14 and S15 are overlaps with TWAS results, which have more than one SNP per gene, and therefore only a gene-level Z-score for association between gene expression and the GWAS trait is reported as produced by SPrediXCan.

- Add the P-value whenever an enrichment of a gene list to a GO term is stated.

We have added this information as requested.

- P7 – "Genes which differed between pre- and post-menarcheal girls were also more likely to be differentially expressed between boys and girls" – those upregulated or downregulated? In the same direction? I would expect boys and pre-menarcheal girls will be more similar, but needs to be shown. Were those genes "differentially expressed between boys and girls" defined and listed before?

We thank the reviewer for this suggestion, which prompted us to think more in depth on the scientific question for this analysis. Because of the age and puberty stage distribution of our samples, the pre-menarcheal girls are not pre-pubertal, therefore, the results cannot be interpreted as suggested by the reviewer. We agree with the reviewer though that without the ability to interpret the direction of the effects in a meaningful way, this statement does not add to the major conclusions of the manuscript. Therefore, we have decided to remove it.

- P7 – "Nine of the genes that changed expression past menarche had also been found to have methylation changes between pre- and post-puberty in girls²². All of these were downregulated past menarche, and six were hypermethylated in girls post puberty (e.g., Fig. 4E)." same as above regarding expected overlap size, and also gene names need to be specified somewhere, not only the one gene in fig 4e.

We have added this information as follows:

"Nine of the genes that changed expression past menarche had also been found to have methylation changes between pre- and post-puberty in females²⁴ (LASP1, GAS7, CTSA, MYL9, AGO4, C1RL, GALNT2, TRAPPC1, C17orf62). This overlap is not higher than expected by chance (Fisher’s test p-value=1)."

- P8 – consider replacing environment with a more appropriate term. Maybe phenotypes?

We have modified the text as follows:

"Previously, we had found that expression of *PGAP1* was modulated by the interaction with genotype across many contexts including cell composition psychosocial factors, such as the extent of self-disclosure of thoughts and feelings, and frequency of verbal arguments within the family"

- Supplementary file 13 not found - http://genome.grid.wayne.edu/puberty/Asthma-TWAS_Blood.csv

We apologize for the oversight. We have provided the correct link now:

http://genome.grid.wayne.edu/puberty/AAM-TWAS_Blood.csv.

- Figure 1C – score range is 0-4, consider matching x tick labels.

Thanks for the suggestion, we have modified the figure as requested.

- The discussion discusses the reason for larger difference in girls, but the results in fig 2a and e show more DEGs in males, or a comparable number, respectively. Please clarify.

We now clarify that the larger differences in gene expression observed in females are across puberty stages, while we find a larger number of DEG in males when considering age.

"While we find a larger number of differentially expressed genes with age in males compared with females, we only detected significant longitudinal gene expression changes in females but not males across puberty stages."

- "longitudinal samples from the same individual processed together in the same batch" – this also means that the first time point was always stored for a ~year more. Can the author comment on an expected effect, or show there is not effect of storage time on samples otherwise similar?

While the two time points for each individual were collected within 1-2 years. The overall study ran for 8 years. Therefore, the distance in collection time between timepoints is expected to have minimal and random effect that will not create a systematic confounding effect with the longitudinal analysis.

- Was there any filtering of the genes done prior to modeling? If so please specify.

We have included this information in the methods section:

"For all gene expression analyses, genes on sex chromosomes and genes with expression below 6 reads or 0.1 counts per million in at least 20% of samples were dropped. "

- How were "all expressed genes in each sex" defined? What is their number?

We have added this information as follows:

"We used the clusterProfiler package for R to run Gene Ontology enrichment analysis for the differentially expressed genes for each variable. We used as background all genes tested for differential expression for that same variable."

The total number of genes tested and therefore used as background varied across variables because different sets of individuals were considered (range: 5224-19358 genes). This information is available in the supplementary tables where we provide all the summary statistics.

- Figure 6 – how was association with asthma risk estimated? Are you sure color in A represents what is stated in the legend and not chromosome number? What do the colors in B stand for?

Association with asthma risk was estimated with our Transcriptome-wide association study as described in the methods. We modified the caption as follows:

"Fig. 7. Changes in gene expression during puberty are associated with age at menarche and asthma risk. A - Miami plot representing our TWAS analysis p-values for age at menarche across the genome (see Methods). Each dot is a gene; red line represents the Bonferroni significance threshold. Genes with higher expression in females undergoing menarche later are plotted above the x axis, genes with higher expression in females undergoing menarche earlier are plotted below the x axis. Shade of gray and x-tick marks separate chromosomes. Green color represents TWAS genes which were also significantly downregulated in postmenarcheal females. B - Expression of genes associated with age at menarche differs across puberty in females. Traits (left column) are connected to their respective differentially expressed genes (middle column) which are connected to the TWAS trait of age at menarche. Color represents the trait: age (grey), puberty (lilac) and menarche (red)."

Reviewer #2 (Remarks to the Author):

Dr Resztak and colleagues have studied changes in gene expression in key processes including immune regulation during pubertal development. The manuscript is well considered and fluently written, and provides some important new insights into the changes in gene expression during pubertal development. I do have some concerns about study design which are detailed below.

Major comments:

The study was conducted in a cohort of patients with asthma, which was presumably a pragmatic decision rather than an intended exploration of gene expression in pubertal development in children with asthma. It remained unclear to me how much the intention of the study was to explore changes in immune system gene expression and how much this was a more global exploration of key genes that are up and downregulated during this developmental period. What impact do the authors feel the particular cohort having asthma might have had on the study outcomes, particularly the assessment of immune system gene expression changes with genetic variation?

We thank the reviewer for this comment as it helped us better frame the focus of the paper. The conclusion we draw about gene expression changes in the immune system during puberty are generalizable to healthy children. Yet, it is known that asthma and other diseases with an immunological component change severity and disease manifestation during puberty. Therefore, we believe that our findings may help explain some of the observed differences in asthma symptoms between males and females before and after puberty. This is the first study exploring these questions and of course our findings should be further studied in larger cohorts. Nevertheless, the disease status of our cohort should have a relatively small impact on the generalizability of our conclusions regarding interactions with genetic variation. To further prove this point, we have performed additional analyses and added new sections to results, methods and discussion where we directly compare gene expression changes associated with both asthma and puberty effects. We also added a new section to the introduction.

Introduction:

"Sex hormones are hypothesized to play a role in the switch of sex skew, as asthma symptoms are also known to worsen around menstruation^{19,20}. Recently, a Mendelian randomization study showed a protective effect of increased sex hormone-binding globulin (SHBG) levels on asthma onset, with a larger effect in females than males²¹. Furthermore, in a cross-sectional study of a large cohort of children with severe asthma, circulating levels of androgens were found to positively associate with improved lung function and symptom control in pubescent males, while late puberty in females was associated with increased estradiol levels and decreased lung function¹⁵ Despite sex differences in susceptibility to asthma before and after puberty onset¹³, earlier onset of puberty has been found to increase the risk of asthma in both sexes¹⁷. "

The title of the new results section is

"Childhood asthma traits and gene expression changes during puberty"

Discussion:

"Previous cross-sectional studies of asthma severity and age in large cohorts have shown that prior to puberty, the incidence of asthma is higher in males, and then post-puberty, adults with asthma are more likely to be female¹⁴. However, a recent longitudinal study throughout adolescence found that asthma severity decreased equally in male and female subjects, probably due to increase in androgens in both sexes⁷⁵. Androgen levels were found to be positively associated with increased pulmonary function^{15,76}, as well as suppression of airway

inflammation^{77,78}. Due to the demographics of our cohort, we captured late pubertal stages in females, which are associated with increased incidence of asthma¹⁴, thus resulting in the sex-specific patterns of gene expression that we observed. Indeed, we found that gene expression changes in late puberty in males are positively correlated with gene expression changes associated with pulmonary function and inversely correlated with changes associated with asthma symptoms, while in females these patterns are reversed. Furthermore, the genes expressed in late pubertal stages in females are enriched for adaptive immunity processes which have been found to be influenced by estrogen⁷³. This is a potential mechanism for previously observed associations between estradiol and decreased pulmonary function in adolescent females with asthma¹⁵ and could be further investigated in conjunction with hormone level measurements."

The definition of the pre- and post groups, in which the comparisons were undertaken were surprising. To include girls that have had menarche in the pre group seems counter-intuitive for a study aimed to detect difference pre and post puberty. The explanations given on which groups were analysed for which experiments were not sufficient – for the ‘Genetic variation interacts with puberty to affect gene expression’ section testing was clearly on puberty stage or menarche; but for the preceding experiments the associations seem much more likely to be due to age than to pubertal stage given that there were late pubertal children included in the first timepoint group.

We apologize for the lack of clarity. In figure 1D and related results, participants assigned to the pre-menarche group had not yet experienced menarche at the time of the study, while those assigned in the post-menarche group are girls who have already started menstruating at the time of the study. We have clarified this in the results as follows:

"We utilized a cross-sectional study design to analyze whether gene expression differed between pre-menarche (females who have not experienced menarche yet at the time of the study, N=22) and post-menarche females (females who are already menstruating at the time of the study, N= 44). A longitudinal analysis was not performed for menarche because this approach was only possible for 4 individuals in our cohort."

Furthermore, to clarify which set of results directly test for an effect of age, we have changed the results title for the relevant section from "Gene expression changes over time in peri-pubertal boys and girls" to "Gene expression changes associated with age".

Minor comments:

Please provide line and page numbering in submitted documents in order to assist reviewing.

We have now provided line and page numbering.

Title – should make reference to the immune system

We have changed the title to "Analysis of transcriptional changes in the immune system associated with pubertal development"

Abstract – ‘change in a sign of’ is not very clear, please rephrase

We removed this sentence as part of the abstract revision requested by one of the reviewers.

Introduction ‘the average age range for puberty...’ this is not very accurate, are we speaking here about age range for onset, mid point of puberty or completion? As per reference <https://doi.org/10.1210/er.2018-00248> ‘In most populations 95 percent of girls experience onset of pubertal development between 8.5 and 13 years and the same percentage of boys between 9 and 13.5 years’

We thank the reviewer for this comment, we have now updated the text to reflect this reference.

"The typical age for puberty onset is approximately 9-13.5 years in males and 8.5-13 years in females, yet there are fundamental differences between males and females in the dynamics of the reactivation of the gonadotropic axis at puberty onset⁴."

Results – Figure 1 – the ‘early puberty’ and ‘late puberty’ labelling of panel A might be confusing and be misinterpreted as ‘children with early ie precocious or late ie delayed puberty’. Perhaps change to ‘Early puberty stage’ or similar

We have modified the labels to "Early puberty stage" and "Late puberty stage".

Figure 1 D – I am surprised that any of the ‘pre’ group have menarche as this is a late event in puberty and should place these girls in the late stage of menarche group. Onset of menarche should be used as an exclusion criterion for being in the ‘pre’ group, otherwise comparisons will not reflect pubertal changes but only those due to age or other factors

We agree with the reviewer and this is indeed what we did. We apologize for the confusion. Participants assigned to the pre-menarche group had not experienced menarche yet at the time of the study, while those assigned in the post-menarche group are girls who have already started menstruating at the time of the study.

We also modified the figure as follows:

Title: Menarche status

X labels: pre-menarche, post-menarche

Results – ‘Mutations near this gene have been implicated in a form of congenital hypogonadism 33–35’ - close chromosomal proximity does not infer an aetiological connection here.

We agree with the reviewer and have removed this sentence from the manuscript.

Discussion – ‘Thus an alternative explanation would be that the gene expression changes associated with pubertal development in boys were very subtle compared to those of girls during the time frame analyzed.’ This seems unlikely given this is a period of major developmental change for both genders.

We agree with the reviewer that puberty is a major developmental stage for both genders, we have rephrased the sentence as follows

"Thus, despite puberty being an important time for developmental changes in both genders, expression changes associated with pubertal development in males during the time frame considered in our dataset were relatively smaller when compared to those of females."

Reviewer #3 (Remarks to the Author):

The study a longitudinal assessment of gene expression by gender over time and is an interesting, novel analysis where they identified genetic effects on gene expression that change during pubertal development and differend by gender and in females with menarche a change in the asthma-associated gene- which may explain changes in females that may increase asthma after puberty and provide mechanistic insights.

Their methods are interesting using a multi-variate adaptive shrinkage approach that uses correlations by gender to improve effect size estimates and power. They also found significant genes associated with age and thymocyte development

Major comments:

1)Overall, the study is interesting, but it is so jam packed with information and it would be helpful to have a clear theme and focus. If they are trying to understand expression related to asthma and how it relates to asthma, that should be listed upfront as main objective. Is there any data on the cohorts as to whether the differentially expressed genes with asthma actually correlated with health outcomes? This would help bring context to the findings.

We thank the reviewer for this comment. We have modified the manuscript in response to this and the other reviewers' similar suggestions. Specifically, we have added a new section to the introduction, we have added new sections to results, methods and discussion where we directly compare gene expression changes associated with both asthma and puberty effects. Additionally, we have revised the abstract (see last comment to this reviewer).

Abstract:

[...] Puberty also marks a shift in sex differences in susceptibility to asthma - males have higher asthma prevalence in childhood, but starting from young adulthood, females are more prone to asthma. [...]

We found that gene expression changes during puberty are correlated with gene expression changes associated with asthma traits. These correlations are in opposite directions for males and females and shed light on potential mechanisms for differences in asthma disease between males and females before and after puberty.

Introduction:

"Sex hormones are hypothesized to play a role in the switch of sex skew, as asthma symptoms are also known to worsen around menstruation^{19,20}. Recently, a Mendelian randomization study showed a protective effect of increased sex hormone-binding globulin (SHBG) levels on asthma onset, with a larger effect in females than males²¹. Furthermore, in a cross-sectional study of a large cohort of children with severe asthma, circulating levels of androgens were found to positively associate with improved lung function and symptom control in pubescent males, while late puberty in females was associated with increased estradiol levels and decreased lung function¹⁵Despite sex differences in susceptibility to asthma before and after puberty onset¹³, earlier onset of puberty has been found to increase the risk of asthma in both sexes¹⁷. "

The title of the new results section is

"Childhood asthma traits and gene expression changes during puberty"

Discussion:

"Previous cross-sectional studies of asthma severity and age in large cohorts have shown that prior to puberty, the incidence of asthma is higher in males, and then post-puberty, adults with asthma are more likely to be female¹⁴. However, a recent longitudinal study throughout adolescence found that asthma severity decreased equally in male and female subjects, probably due to increase in androgens in both sexes⁷⁵. Androgen levels were found to be positively associated with increased pulmonary function^{15,76}, as well as suppression of airway inflammation^{77,78}. Due to the demographics of our cohort, we captured late pubertal stages in females, which are associated with increased incidence of asthma¹⁴, thus resulting in the sex-specific patterns of gene expression that we observed. Indeed, we found that gene expression changes in late puberty in males are positively correlated with gene expression changes associated with pulmonary function and inversely correlated with changes associated with asthma symptoms, while in females these patterns are reversed. Furthermore, the genes expressed in late pubertal stages in females are enriched for adaptive immunity processes which have been found to be influenced by estrogen⁷³. This is a potential mechanism for previously observed associations between estradiol and decreased pulmonary function in adolescent females with asthma¹⁵ and could be further investigated in conjunction with hormone level measurements."

2) I would definite have a bioninformatic/statistician review the methods. It appears there are so many comparisons, it would be difficult to know if the approach is solid.

We strengthened the investigative team by adding our long-term collaborator and biostatistician Xiaoquan Wen, who reviewed all the statistical methods used.

3) Clarity on the meanings of the interactions between genotype and puberty should be described.

We added a paragraph to clarify the meaning and biological relevance of interactions between genotype and puberty.

"We sought to identify genetic variants whose effect on gene expression changed throughout puberty. For example, an allele may increase gene expression in early puberty but decrease it in late puberty (qualitative change in the genetic effect on gene expression during puberty). A quantitative change in the genetic effect on gene expression during puberty would be defined instead as an allele that is associated with increased gene expression but the magnitude of the increase varies in different puberty stages. This type of analysis is also known as dynamic or context quantitative trait locus (QTL) mapping."

4) There are a couple of references missing that should be discussed and described in the context of how these related to clinical findings as relates to age and pubertal changes in asthma.

Severe asthma during childhood and adolescence: A longitudinal study.

Ross KR, Gupta R, DeBoer MD, Zein J, Phillips BR, Mauger DT, Li C, Myers RE, Phipatanakul W, Fitzpatrick AM, Ly NP, Bacharier LB, Jackson DJ, Celedón JC, Larkin A, Israel E, Levy B, Fahy JV, Castro M, Bleecker ER, Meyers D, Moore WC, Wenzel SE, Jarjour NN, Erzurum SC, Teague WG, Gaston B.J Allergy Clin Immunol. 2020 Jan;145(1):140-146.e9. doi: 10.1016/j.jaci.2019.09.030. Epub 2019 Oct 14.PMID: 31622688 Clinical Trial.

Effects of endogenous **sex hormones** on lung function and symptom control in adolescents with asthma.

DeBoer MD, Phillips BR, Mauger DT, Zein J, Erzurum SC, Fitzpatrick AM, Gaston BM, Myers R, Ross KR, Chmiel J, Lee MJ, Fahy JV, Peters M, Ly NP, Wenzel SE, Fajt ML, Holguin F, Moore WC, Peters SP, Meyers D, Bleecker ER, Castro M, Coverstone AM, Bacharier LB, Jarjour NN, Sorkness RL, Ramratnam S, Irani AM, Israel E, Levy B, Phipatanakul W, Gaffin JM, Gerald Teague W.BMC Pulm Med. 2018 Apr 10;18(1):58. doi: 10.1186/s12890-018-0612-x.PMID: 29631584 Free PMC article.

Baseline Features of the Severe Asthma Research Program (SARP III) Cohort: Differences with Age.

Teague WG, Phillips BR, Fahy JV, Wenzel SE, Fitzpatrick AM, Moore WC, Hastie AT, Bleecker ER, Meyers DA, Peters SP, Castro M, Coverstone AM, Bacharier LB, Ly NP, Peters MC, Denlinger LC, Ramratnam S, Sorkness RL, Gaston BM, Erzurum SC, Comhair SAA, Myers RE, Zein J, DeBoer MD, Irani AM, Israel E, Levy B, Cardet JC, Phipatanakul W, Gaffin JM, Holguin F, Fajt ML, Aujla SJ, Mauger DT, Jarjour NN.J Allergy Clin Immunol Pract. 2018 Mar-Apr;6(2):545-554.e4. doi: 10.1016/j.jaip.2017.05.032. Epub 2017 Aug 31.PMID: 28866107 Free PMC article.

This could be modified depending on the focus and direction of the paper.

We thank the reviewer for the suggestion, we have now increased the focus on asthma, with new analyses, and also added these references to the discussion (see response to comment 1, Reviewer 3).

5) The discussion of reprogramming and changes and consideration of why it explains autoimmune disease seems that there are a few themes being described from the findings and it is confusing

We thank the reviewer for this comment. Following the advice to focus the paper on asthma, we have removed this section and instead expanded sections on asthma both in the introduction and discussion (see response to the first major comment by this reviewer).

6) if the entire cohort had asthma- it would be helpful to see if any of the findings held true in a non asthma cohort

We agree with the reviewer that ideally we would like to conduct this study in a cohort of children without asthma, however we do not have access to such a cohort and we are not aware of such a dataset being available. We have now refined the focus of the manuscript on asthma and changes occurring pre/post-puberty in male and female patients. See response to comment 1, Reviewer 3.

7) how was asthma defined in the cohort. Many basic details are missing.

We apologize for the lack of details. We have now added this information to the methods.

"To be included in the study, youth were required to be between 10 and 17 years of age at the time of recruitment and diagnosed with at least mild to persistent asthma by a physician (with diagnosis confirmed from medical records)."

8) how was puberty assessed. Were there trained clinicians on Tanner staging? Were hormonal levels ascertained as well?

We have clarified puberty assessment in the methods as below:

"Puberty staging was self-reported as this has been shown to be reliable, less intrusive, and allows for large epidemiological studies (Chavarro et al. 2017; Balzer et al. 2019). Pubertal development was assessed at each timepoint prior to the blood draw for gene expression measurement via a sex-specific self-report questionnaire (Carskadon, 1993). The questions and possible responses can be found in Table S2. The overall pubertal development score is a mean of the responses to specific questions 1-5, given that at least four valid responses are provided. Pubertal development stages are analogous to the Tanner stages (Marshall, 1969, Marshall, 1970)."

Hormonal levels were not measured in this cohort.

Minor comments:

The abstract doesn't fit a normal structure and has examples that again detract from the paper. What is the overall purpose—It seems it is trying to analyze changes in puberty and come with all types of associations and relationship with diseases without a clearly driven hypothesis or approach of questions they want to answer.

Following the reviewer's advice, we have revised the abstract to reflect the main theme of this revised manuscript on puberty, immune cell gene expression and asthma, and incorporating the new results.

REVIEWERS' COMMENTS

Reviewer #1 (Remarks to the Author):

Authors have satisfactorily revised the manuscript. The fact that no similar dataset exists for healthy people makes this dataset and its analysis even more important, as a first glimpse into the changes of the human immune system during puberty. However, as there is no way of estimating how much that glimpse is unique to asthma, the word asthma is better be added at the title as well.

Reviewer #2 (Remarks to the Author):

The authors have made major revisions to the manuscript and have appropriately responded to all of the comments that I have made. The paper is now improved, and is a valuable data source - although it remains a somewhat unclear on its conclusions on the relationship between puberty, gender, the immune system and asthma across puberty.

Reviewer #3 (Remarks to the Author):

appropriately addressed

We thank the reviewers for their insightful comments and suggestions. Reviewers 2 and 3 indicated that they are satisfied with the changes made. Below is our response to reviewer 1 (in blue).

Reviewer #1 (Remarks to the Author):

Authors have satisfactorily revised the manuscript. The fact that no similar dataset exists for healthy people makes this dataset and its analysis even more important, as a first glimpse into the changes of the human immune system during puberty. However, as there is no way of estimating how much that glimpse is unique to asthma, the word asthma is better be added at the title as well.

We have modified the title to include the word “asthma”, as suggested both by the reviewer and the editor. The new title “Analysis of transcriptional changes in the immune system associated with pubertal development in a longitudinal cohort of children with asthma”.